# Adapted-Language ViT: Empowering Self-Supervised Vision Transformers with LLMs

## Abstract

The integration of Large Language Model (LLMs) blocks with Vision Transformers (ViTs) holds immense promise for vision-only tasks by leveraging the rich semantic knowledge and reasoning capabilities of LLMs. However, a fundamental challenge lies in the inherent modality mismatch between text-centric pretraining of LLMs and vision-centric training of ViTs. Direct fusion often fails to fully exploit the LLM's potential and suffers from unstable finetuning. As a result, LLM blocks are kept frozen while only the vision components are learned. As a remedy to these challenges, we introduce Adapted-Language Vision Transformers (ALViT), a novel approach that bridges this modality mismatch through a synergistic pre-training strategy. ALViT co-adapts a ViT backbone and an LLM fusion block by (1) employing Masked Auto-Encoding (MAE) to pre-train the ViT for richer visual representations, and (2) concurrently training Low-Rank Adaptation (LoRA) layers within the LLM block using the MAE objective. This joint optimization guides the ViT to produce LLM-aligned features and the LLM to effectively interpret visual information. We demonstrate through extensive experiments that ALViT significantly improves performance in various downstream vision tasks, showcasing an effective and efficient way to harness LLM knowledge for visual understanding.

## 1 Introduction

The remarkable success of Large Language Models (LLMs) (Brown et al., 2020; Touvron et al., 2023a) has revolutionized natural language processing, demonstrating advanced capabilities in understanding, generation, and reasoning. This success has led to significant interest in extending their power to other modalities, particularly vision, impacting to the field of Vision-Language Models (VLMs) (Radford et al., 2021; Alayrac et al., 2022; Tschannen et al., 2025). A promising direction within VLMs involves directly integrating powerful pre-trained LLM components with Vision Transformer (ViT) (Dosovitskiy et al., 2020) backbones, aiming to fuse visual models with the extensive semantic knowledge and reasoning abilities learned by LLMs from vast textual corpora.

However, these applications of LLM for vision explore them in a generative framework, limiting their application to discriminative computer vision tasks. Pioneering works like LM4Vision (Pang et al., 2023) have explored fusing ViT features with terminal blocks of LLMs while learning a computer vision task, hinting at the potential benefits. Regardless, a critical hurdle persists: the alignment of representations originating from different modalities. LLMs are pre-trained exclusively on text, optimizing their internal representations for linguistic structures and concepts. Similarly, ViTs learn visual features optimized for tasks like image recognition. Simply injecting visual features into a text-centric LLM block often results in suboptimal alignment (Liang et al., 2022), where the LLM struggles to effectively ground its textual knowledge in the visual domain. Furthermore, adapting the large LLM component to the visual modality by joint fine-tuning can be computationally prohibitive and risks catastrophic forgetting or training instabilities (Pang et al., 2023; Lai et al., 2024).

To address these challenges, we introduce **A**dapted-**L**anguage **Vi**sion **T**ransformers (ALViT), a novel framework designed to foster a more profound and efficient synergy between ViTs and LLMs for discriminative vision tasks. Our core idea is a two-fold strategy:

1. **Enhanced Visual Representation Learning:** We pre-train the ViT backbone using Masked Auto-Encoding (MAE) (He et al., 2022). This self-supervised objective encourages the ViT

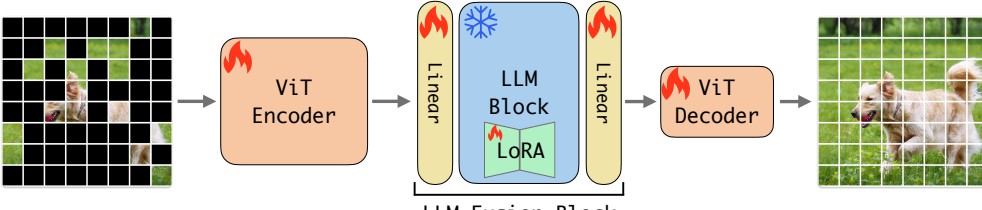

Figure 1: Architecture diagram of our **Adapted-Language Vision Transformer (ALViT)**. Input image patches are processed by the ViT Encoder. The resulting visual features are then passed through an LLM Fusion Block (comprising linear projections and an LLM transformer block adapted with LoRA). For MAE pre-training, a lightweight decoder reconstructs masked patches. For fine-tuning, the decoder is removed, and a task-specific head is added.

> to learn richer, more context-aware visual representations that we hypothesize are more informative for the LLM blocks.

2. **Efficient LLM Adaptation and Modality Bridging:** Simultaneously, we adapt the fused LLM block (e.g., from LLaMA) using Low-Rank Adaptation (LoRA) (Hu et al., 2022). Crucially, these LoRA layers are trained *concurrently* with the MAE pre-training of the ViT, using the same MAE reconstruction loss. This joint optimization allows the LLM to efficiently learn to interpret the evolving visual features, effectively translating its vast semantic knowledge to the visual domain without requiring full fine-tuning of the LLM.

This synergistic pre-training process is key: the ViT learns to leverage the pretrained LLM representations, while the LLM (via LoRA) learns to "understand" these visual features, thereby bridging the modality mismatch from both ends. Our contributions are fourfold:

- We propose ALViT, a novel architecture and pre-training strategy that co-adapts a ViT and an LLM block through joint MAE-based self-supervision and LoRA-based LLM adaptation, effectively mitigating the alignment issue between representations of different modalities.

- We demonstrate that this concurrent optimization of LoRA layers within the LLM during MAE pre-training enables efficient and stable adaptation of the LLM, allowing it to effectively leverage its textual knowledge for visual understanding.

- We show through extensive experiments on benchmark computer vision tasks that ALViT significantly outperforms existing LLM-fusion approaches that employ more direct strategies, pushing forward leveraging frozen LLM capabilities in vision models.

- We provide intriguing analyses regarding the attention entropies of ALViT and how it achieves stronger performance through improved background robustness.

## 2 BACKGROUND AND RELATED WORK

**Self-supervised learning.** Self-supervised learning (SSL) has emerged as a powerful paradigm for leveraging readily available unlabeled data. SSL methods have achieved widespread success in the broader machine learning community, starting with earlier contrastive approaches (Chen et al., 2020b; He et al., 2020), achieving new frontiers in representation learning otherwise unreachable with full-supervised techniques. More recently, SSL approaches have powered foundation models in a wide range of domains, from NLP (Touvron et al., 2023a;b; Devlin et al., 2019) to vision (Caron et al., 2021; Grill et al., 2020; Naeem et al., 2024).

**Masked image modeling.** Masked image modeling is an established example of self-supervised learning methods for computer vision, initially pioneered by stacked denoising autoencoders (Vincent et al., 2010). Motivated by the success of masked language modeling approach of BERT (Devlin et al., 2019), a plethora of follow-up works proposed novel self-supervised masked image modeling techniques (Chen et al., 2020a; Bao et al., 2021; Zhou et al., 2021; Dosovitskiy et al., 2020). Among these works, Masked Auto-Encoders (MAE) (He et al., 2022) stand out with their accelerated pretraining approach consisting of a heavyweight encoder observing only a small fraction of image

patches and a lightweight decoder reconstructing the original image features. MAE has established itself as a strong approach not only for global image recognition but also for more challenging fine-grained visual recognition tasks, such as object detection (Li et al., 2022b).

**Using frozen LLM blocks for visual tasks.** Closest to our work are the works directly employing frozen pretrained LLM blocks with vision transformers (Pang et al., 2023; Lai et al., 2024; Bai et al., 2025). Among these, Pang et al. (2023) is the pioneering work that showed that using frozen LLM blocks on top of vision transformers can provide strong performance gains on pure vision tasks. However, Pang et al. (2023) did not aim to achieve *state-of-the-art* performance on visual recognition but rather provided relative performance improvement on a wide range of vision tasks.

In this work, we combine the powers of self-supervised learning, the initial explorations of Pang et al. (2023), and LoRA adaptations together to achieve significantly improved downstream performance, differing from the previous art. Evidenced by our experiments, our work provides stronger recipes for achieving stellar visual recognition performance while effectively leveraging the LLM blocks.

## 3 ALViT: Adapted-Language Vision Transformers

While LM4Vision (Pang et al., 2023) demonstrated the potential of fusing Vision Transformers (ViTs) with the terminal block of a Large Language Model (LLM), the direct introduction of this transformer block introduces a modality mismatch due to the LLM's text-centric pre-training and Vision Transformer's visual processing. To address this, we propose a two-fold strategy. First, we introduce Self-Supervised Learning (SSL) using Masked Auto-Encoding (MAE) during the pre-training of the ViT backbone. This step aims to better align visual representations with the language modality. Second, to adapt the LLM component (e.g., LLaMA), pre-trained solely on text, we incorporate Low-Rank Adaptation (LoRA). This allows the LLM to efficiently translate its extensive semantic knowledge, learned from billion-scale textual data, to the visual domain, thereby improving performance on target computer vision tasks.

### 3.1 ALViT: Adapted-Language Vision Transformer

We introduce **A**dapted-**L**anguage **Vi**sion Transformers (ALViT), with the aim of effectively bridging the the representation alignment issue between vision and language representations when using language trained transformer blocks in vision transformers. The core intuition is to enable a synergistic co-adaptation: the ViT learns to produce visual features amenable to language processing, while the LLM block learns to interpret these visual features, all within a unified pre-training framework.

Our ALViT architecture (illustrated in Figure 1) comprises of three main components:

1. **Vision Transformer (ViT) Encoder ($\mathbf{M}_{Enc}$):** Following (Dosovitskiy et al., 2020), the standard ViT maps input patches $x$ into latent visual representations $z_v = \mathbf{M}_{Enc}(x)$.

2. **LLM Fusion Block ($\mathbf{M}_{LLM}^{\text{fuse}}$):** This module integrates a pre-trained LLM transformer block (e.g., from LLaMA (Touvron et al., 2023a)) into pipeline to enrich the visual features $z_v$. To manage differing hidden dimensions and facilitate adaptation, $z_v$ is first projected by a linear layer $\mathbf{M}_L^1$, then processed by the LLM block $\mathbf{M}_{LLM}$, and finally projected back by $\mathbf{M}_L^2$. Thus, the enhanced latent features are $z_v' = \mathbf{M}_L^2 \cdot \mathbf{M}_{LLM} \cdot \mathbf{M}_L^1(z_v)$. We denote this entire compound mapping as $\mathbf{M}_{LLM}^{\text{fuse}}(z_v) \rightarrow z_v'$.

3. **Lightweight MAE Decoder ($\mathbf{M}_{Dec}$):** For self-supervised pre-training, a shallow transformer decoder, similar to (He et al., 2022), takes the enhanced latent features $z_v'$ from visible patches and reconstructs the original masked image patches $x'$.

The complete pre-training pipeline for an input image $x$ can thus be expressed as:

$$x' = \mathbf{M}_{Dec}\left(\mathbf{M}_{LLM}^{\text{fuse}}\left(\mathbf{M}_{Enc}(x_{\text{vis}})\right), x_{\text{mask\_ids}}\right), \qquad (1)$$

where $x_{\text{vis}}$ represents visible patches fed to the encoder, and $x_{\text{mask\_ids}}$ represents information about the masked patches required by the decoder for reconstruction (e.g., their positional encodings).

### 3.2 SYNERGISTIC PRE-TRAINING FOR MODALITY ALIGNMENT

The core component of ALViT is its pre-training strategy, designed to address the modality mismatch through self-supervised pretraining. This involves concurrently training the ViT via Masked Auto-Encoding (MAE) and adapting the LLM fusion block using LoRA.

#### 3.2.1 SELF-SUPERVISED VISUAL REPRESENTATION LEARNING VIA MAE

**Intuition.** Standard ViT training (e.g., on ImageNet) learns features optimized for classification but these features often fail to capture deeper semantics required for other computer vision tasks (He et al., 2022). However, self-supervised pretrained backbones learn more generic features often directly usable across a plethora of computer vision tasks (Oquab et al., 2023; He et al., 2022). We utilize Masked Auto-Encoding (MAE) (He et al., 2022) as the self-supervision framework owing to its recent success in learning robust features and its efficiency (He et al., 2022; Tschannen et al., 2025). MAE learns holistic and context-aware representations by reconstructing heavily masked inputs. When learned together with a LLM block, we hypothesize that such representations are inherently richer and more compatible with the high-level understanding capabilities of LLM block.

**Mechanism.** We follow the standard MAE pre-training strategy proposed by (He et al., 2022). An input image $x$ is divided into $N$ non-overlapping patches. A high percentage (e.g., 75%) of these patches are randomly masked out. Only the visible patches $x_{\text{vis}}$ are processed by the ViT encoder $\mathbf{M}_{Enc}$ and subsequently by the LLM fusion block $\mathbf{M}_{LLM}^{\text{fuse}}$. The lightweight decoder $\mathbf{M}_{Dec}$ takes the output from the LLM block and reconstructs the original pixels of the masked patches from the enhanced latent representations $z_v'$ and the positional embeddings of all patches. The learning objective minimizes the Mean Squared Error (MSE) between the reconstructed and original masked patches. This process trains the ViT backbone $\mathbf{M}_{Enc}$.

#### 3.2.2 EFFICIENT LLM ADAPTATION WITH LOW-RANK ADAPTATION (LoRA)

**Intuition.** Pre-trained LLMs possess vast world knowledge and complex reasoning abilities encoded in their weights. Fine-tuning the entire LLM for a vision task is computationally prohibitive and risks catastrophic forgetting of its semantic understanding capabilities that we want to utilize for visual understanding. LoRA (Hu et al., 2022) offers a parameter-efficient solution, allowing us to "steer" the LLM's knowledge towards the visual domain by training only a small number of additional parameters. It also allows for stable finetuning of the LLM block without the risk of the larger LLM block collapsing the training signal.

**Mechanism.** We inject LoRA layers into the query ($W_q$) and value ($W_v$) projection matrices of the LLM block $\mathbf{M}_{LLM}$. For a pre-trained weight matrix $W_0 \in \mathbb{R}^{d \times k}$, its update is represented by a low-rank decomposition $W_0 + \Delta W = W_0 + BA$, where $B \in \mathbb{R}^{d \times r}$, $A \in \mathbb{R}^{r \times k}$, and the rank $r \ll \min(d, k)$. Only $A$ and $B$ are trainable. The original LLM weights $W_0$ remain frozen keeping their pre-trained knowledge secure.

#### 3.2.3 JOINT OPTIMIZATION: THE KEY TO MODALITY BRIDGING

A critical aspect of our method is that the LoRA layers within $\mathbf{M}_{LLM}^{\text{fuse}}$ are trained *concurrently* with the ViT backbone during the MAE pre-training phase. The MAE reconstruction loss not only guides the ViT but also backpropagates through the LLM fusion block, updating the LoRA parameters. This joint optimization fosters a synergistic co-adaptation during learning, while the ViT ($\mathbf{M}_{Enc}$) learns to produce visual embeddings that are not only good for reconstruction but are also effectively processed and enhanced by the LLM block. The LLM block learns to interpret and refine these evolving visual embeddings via LoRA in $\mathbf{M}_{LLM}$, leveraging its pre-trained frozen textual knowledge to enhance them with richer semantics relevant to the visual context.

This simultaneous learning process is crucial for bridging the modality mismatch, as it forces the two modalities to be jointly aligned rather than adapting one to a fixed representation of the other. The LLM is not just passively processing ViT features; it is actively being aligned to understand the visual world while the ViT learns to present this information in a more digestible format in the LLM space.

### 3.3 ARCHITECTURAL ADJUSTMENTS FOR CROSS-MODAL LLM PROCESSING

To further enhance the LLM block's suitability for processing visual information, we incorporate specific architectural modifications, following the existing works (Pang et al., 2023; Lai et al.,

2024). **(1) Bidirectional Attention.** LLMs commonly use causal attention masks. However, visual information in an image does not possess sequential causality the same way as language. Thus, we replace the causal attention mechanism in the LLM block with bidirectional attention. This allows each visual token representation in the LLM block to attend to all others, allowing for a holistic understanding. **(2) Removal of Rotary Pos. Embeddings (RoPE).** RoPE (Su et al., 2024), commonly used in LLMs, encodes absolute and relative positional information tailored for text sequences. Since our ViT backbone already incorporates learned positional embeddings for visual patches, and the nature of spatial relationships in images differs from sequential text, we remove RoPE from the LLM block. This simplifies the architecture, prevents the imposition of text-specific biases onto visual features, and ensures consistency with typical ViTs.

### 3.4 Downstream Fine-tuning

After the MAE-based pre-training with joint LoRA adaptation, ALViT is fine-tuned for specific downstream computer vision tasks (e.g., image classification). For fine-tuning, we discard the MAE decoder ($\mathbf{M}_{Dec}$), and add a task-specific head (e.g., a linear classifier) on top of the output features $z'_v$. During fine-tuning, the ViT backbone, the linear projection layers $\mathbf{M}^1_L, \mathbf{M}^2_L$, and the LoRA parameters within the LLM block can be further trained. The original weights of the LLM block $\mathbf{M}_{LLM}$ remain frozen, preserving its extensive learned knowledge while allowing targeted adaptation through LoRA. This strategy ensures efficient transfer of learned representations to downstream tasks.

## 4 Experiments

We now discuss our experiments and highlight the strengths of our Adapted-Language Vision Transformers (ALViT).

**Datasets.** For our image classification experiments, we utilize the ImageNet-1K training and validation splits (Deng et al., 2009). In addition, we report evaluation results on several domain-shift benchmarks, namely ImageNet-C (Hendrycks & Dietterich, 2019), ImageNet-A (Hendrycks et al., 2021b), ImageNet-SK (Wang et al., 2019), ImageNet-V2 (Recht et al., 2019), and ImageNet-R (Hendrycks et al., 2021a). Furthermore, we report additional results on ImageNet-9 benchmark (Xiao et al., 2020), which measures the reliance of a model on background and foreground features. Among its splits, we choose the *mixed same* and the *mixed random*. In the former, backgrounds of images are randomly replaced with the background of another image of the same class, and in the latter the background is replaced with the background of an image of a completely random class. Finally, for our fine-grained visual recognition experiments, we use the MS COCO (Lin et al., 2014) object detection dataset. We report our results on the COCO *val* set, following Li et al. (2022b). We refer the reader to Section B.1 for the results, where we show that ALViT obtains similar improvements.

**Pretraining.** Our pre-training settings closely mirror that of the original MAE work He et al. (2022), including all of the hyperparameters related to the training (learning rate, mask ratio, *etc.*). We pre-train both vanilla MAE-ViT baselines and our ALViT for a total of 800 epochs, following He et al. (2022). For our LLM block, unless otherwise specified, we utilize the $32^{nd}$ transformer block of LLaMA 1 Touvron et al. (2023a), following our ablations in Section 4.2. As described in Section 3, while the original LLM weights are always kept frozen, we also integrate LoRA (Hu et al., 2022) into the $Q$ and $V$ projection matrices, both with a rank of 16, constituting a very small fraction (0.3%) of the number of trainable parameters.

**End-to-end Finetuning.** For image classification, we perform finetuning for 100 epochs on both the baselines and our ALViT after the pre-training stage, while adhering to all of the hyperparameter settings and other training details presented in (He et al., 2022). For fine-grained visual recognition, we train both the baselines ALViT for 100 epochs after pre-training, while adhering to all training settings in ViTDet (Li et al., 2022b). Finally, with the exception of Tables 1 & 2), all results are reported with a random seed of 0, following our baselines.

### 4.1 Image Classification

We evaluate ALViT on ImageNet-1K benchmark and its variants designed to test robustness to domain shifts (IN-A, IN-SK, IN-V2, IN-R) and common corruptions (IN-C). Furthermore, we replicated the results of Pang et al. (2023) using the authors' code to obtain the standard deviation, as they were

Table 1: ALViT achieves significantly better Top-1 accuracy (%) in frozen LLM augmented model setting on IN-1K, drastically improving over the supervised baselines. We also demonstrate significantly enhanced robustness across challenging variants (IN-A, IN-SK, IN-V2, IN-R, IN-C). ALViT consistently outperforms both supervised baselines and the strong MAE-pretrained ViT/B. Each result denotes the average of 3 random seeds along with associated standard deviations. To obtain the standard deviations, we reproduced the results of Pang et al. (2023), and provide the original numbers in gray for reference. **Bold** indicates the best result.

| Training | Model | IN-1K | IN-A | IN-SK | IN-V2 | IN-R | IN-C |
|---|---|---|---|---|---|---|---|
| Supervised-Only [LM4Vision] (Pang et al., 2023) | ViT/B* | 80.6 | 23.4 | 31.9 | – | 43.5 | 60.2 |
| | ViT/B+LM1* | 81.7 | 26.9 | 33.2 | – | 44.3 | 62.1 |
| | ViT/B | $79.58_{\pm0.81}$ | $22.78_{\pm3.77}$ | $30.61_{\pm0.68}$ | $67.48_{\pm1.07}$ | $42.57_{\pm1.42}$ | $59.73_{\pm1.69}$ |
| | ViT/B+LM1 | $80.50_{\pm0.25}$ | $23.22_{\pm0.80}$ | $31.06_{\pm0.48}$ | $68.69_{\pm0.41}$ | $41.92_{\pm0.57}$ | $61.24_{\pm0.27}$ |
| MAE Pretrained | ViT/B | $83.11_{\pm0.09}$ | $33.64_{\pm0.11}$ | $35.69_{\pm0.30}$ | $72.73_{\pm0.21}$ | $49.88_{\pm0.32}$ | $62.86_{\pm0.01}$ |
| | ALViT/B *(Ours)* | $\mathbf{83.63_{\pm0.04}}$ | $\mathbf{36.39_{\pm0.28}}$ | $\mathbf{36.36_{\pm0.61}}$ | $\mathbf{73.15_{\pm0.02}}$ | $\mathbf{50.17_{\pm0.16}}$ | $\mathbf{63.44_{\pm0.05}}$ |
| | | +0.52 | +2.75 | +0.67 | +0.42 | +0.29 | +0.58 |

reported only with a single seed. The results in Table 1 demonstrate the performance improvements of our ALViT.

**ALViT Outperforms All Baselines.** Our ALViT/B model significantly outperforms previous works, achieving $\mathbf{83.63}\%$ top-1 accuracy. This surpasses not only the supervised ViT/B (79.58%) but also the prior LLM-augmented supervised model, LM4Vision (Pang et al., 2023) (80.50%). More importantly, ALViT outperforms the strong MAE-ViT/B baseline (83.11%), demonstrating the impact of our synergistic LLM integration beyond standard MAE pretraining. We note that ALViT is much more stable between random seeds compared to LM4Vision Pang et al. (2023), as quantified by the standard deviation between multiple runs. We hypothesize that the gap between our reproductions and the values reported in Pang et al. (2023) could be attributed to these instabilities.

**ALViT Better Leverages LLM Benefits.** The MAE-pretrained ViT/B already provides a powerful visual backbone, outperforming the supervised ViT/B+LM1 (83.11% vs. 80.50% on IN-1K). However, ALViT builds on this strong foundation consistently and achieves respectable improvements. The improvements of ALViT over the MAE-ViT baseline (e.g., +0.52% on IN-1K, +2.75% on IN-A) directly validate our hypothesis: concurrently training the LoRA-adapted LLM block during MAE pre-training enables the LLM to effectively process and enhance visual features. This joint optimization better bridges the modality mismatch, allowing the LLM to contribute semantic knowledge to the visual task, a benefit not realized by simply pre-training the ViT with MAE alone or even with extra capacity as shown in Section 4.2.

**Enhanced Robustness and Generalization.** The advantages of ALViT become even more pronounced on robustness benchmarks. On IN-A, a particularly challenging adversarial dataset, ALViT achieves a $\mathbf{13.24}\%$ improvement over LM4Vision (Pang et al., 2023) and a $\mathbf{2.75}\%$ over the MAE-ViT baseline. ALViT also attains respectable gains over both LM4Vision (Pang et al., 2023) and MAE ViT baselines on IN-SK (+3.13%, +0.67%), IN-C (+2.20%, +0.58%), and IN-V2 (+4.46%, +0.42%).

With these results, the superior performance over the MAE-pretrained ViT shows that our method of integrating and adapting the LLM component brings tangible benefits beyond self-supervised visual pre-training. Second, it shows substantial improvements on robustness benchmarks (especially IN-A). The results indicate that ALViT successfully leverages the LLM's knowledge to achieve improved resilience against out-of-distribution samples which is particularly important for real-world vision systems. Third, by outperforming previous attempts at LLM-ViT fusion, like LM4Vision (Pang et al., 2023), ALViT demonstrates the importance of both a strong pre-training paradigm (MAE) and an efficient adaptation strategy (concurrent LoRA training) to reap the benefits of the LLM block.

## 4.2 ABLATIONS

In this section, we quantify the importance of the several building blocks of our approach: the pre-trained LLM representations, the importance of LoRA and their combination with MAE pretraining. We ablate these components and report the results on Tables 2 & 3 on ImageNet-1K. We present

Table 2: Ablation analysis of ALViT components on ImageNet-1K confirms that ALViT design choices are essential to achieve the best performance. "Trainable Params" refers to parameters updated during the final fine-tuning stage, which includes the entire ViT, projections, and LoRA if present). ViT/B+MLP models are configured to match the trainable parameters of corresponding LLM-augmented models. Models are MAE pretrained and each result is the average of 3 random seeds with associated standard deviations. **Bold** indicates the best result.

| | Model | Trainable Params. | IN-1K |
|---|---|---|---|
| **(a)** | ViT/B | 86.8M | $83.11_{\pm0.09}$ |
| **(b)** | ViT/B+MLP-Proj. Match | 92.9M | $83.13_{\pm0.06}$ |
| **(c)** | ViT/B+LM1 | 92.9M | $83.13_{\pm0.02}$ |
| **(d)** | ViT/B+MLP-LoRA Match | 93.1M | $83.21_{\pm0.11}$ |
| **(e)** | ViT/B+Random LM1+LoRA | 93.1M | $83.25_{\pm0.09}$ |
| **(f)** | ALViT/B *(Ours)* | 93.1M | $\mathbf{83.63_{\pm0.04}}$ |

Table 3: Ablation analysis with different LLaMA 1 blocks and different LLMs' final blocks (LLaMA 1 7B (Touvron et al., 2023a), Gemma 2 9B (Team et al., 2024), LLaMA 3.1 8B (Grattafiori et al., 2024), rows show that ALViT's improvements hold across different LLMs, with increasing improvements towards the final blocks. All experiments had a random seed of 0. **Bold** indicates the best results.

| | | LLM Type | Block | Trainable Params. | IN-1K |
|---|---|---|---|---|---|
| MAE ViT/B | | N/A | N/A | 86.8M | 83.2 |
| | **(a)** | LLaMA 1 | 1 | 93.1M | 83.2 |
| | **(b)** | LLaMA 1 | 16 | 93.1M | 83.4 |
| | **(c)** | LLaMA 1 | 31 | 93.1M | 83.5 |
| ALViT/B | **(d)** | LLaMA 1 *(default)* | 32 | 93.1M | **83.6** |
| | **(e)** | Gemma 2 | 42 | 93.1M | 83.5 |
| | **(f)** | LLaMA 3.1 | 32 | 93.1M | **83.6** |
| | **(g)** | LLaMA 3.1-Instruction | 32 | 93.1M | **83.6** |

more results in Section B.2 on how ALViT performs better compared to a baseline with a trainable LLM block and an ablation with $> 1$ LLM blocks.

**LoRA Adaptation is Crucial for Leveraging LLM Benefits with MAE Pre-training.** Comparing row (a) and (c) of Table 2, we observe that the frozen LLM variant without any LoRA fine-tuning in row (c) (83.13% IN-1K) achieves on-par performance with the baseline MAE ViT of row (a) (row a: 83.11% IN-1K). Without adaptation, the LLM block does not benefit from the richer features coming from the MAE-pre-training. This is in contrast with Pang et al. (2023) where the improvements were possible without LoRA on a weaker baseline. However, when we introduce LoRA and adapt the LLM block, as in our full ALViT/B model (row f), performance significantly improves to **83.63%** on IN-1K. This is a clear improvement over both the MAE ViT/B baseline (row a) and the frozen LLM variant without LoRA (row c). Coupled with our study on multiple random seeds in Table 1, these results confirm that LoRA-based adaptation is *essential* for effectively bridging the modality mismatch and allowing the LLM to use enhanced visual representations.

**ALViT's Gains are Not Merely from Increased Parameters.** A critical question is whether ALViT's improvements stem from our model design or from an increased number of trainable parameters introduced by the linear projections and LoRA. To investigate this, we report additional results with two stronger baselines in Table 2, namely **(1) ViT/B+MLP (Proj. Match, row b)** and **(2) ViT/B+MLP (LoRA Match, row d)**. The former's total trainable parameters (92.9M) match those of the ViT/B+LM1 (row c), which includes the ViT and the trainable linear projections, whereas the latter's total trainable parameters (93.1M) match those of our full ALViT/B model (row f), which includes the ViT, trainable projections, and trainable LoRA layers.

Comparing row (b) with row (c), the ViT/B+MLP (Proj. Match) performs on-par on IN-1K compared to the frozen LLM without LoRA. However, the crucial comparison is between our full ALViT/B model (row f) and its parameter-matched MLP counterpart (row d). ALViT/B achieves **83.63%** on IN-1K, outperforming ViT/B+MLP (LoRA Match) (row d: 83.21% IN-1K) by $+0.42\%$ on IN-1K.

| | ViT/B | | ALViT/B (Ours) | |
|---|---|---|---|---|
| | Attn. Entropy | Patch Norm | Attn. Entropy | Patch Norm |

Figure 2: ALViT simultaneously exhibits lower attention entropies and higher patch norms for informative foreground regions across the images. Brighter colors indicate patches with high attention entropy and/or patches with higher norm.

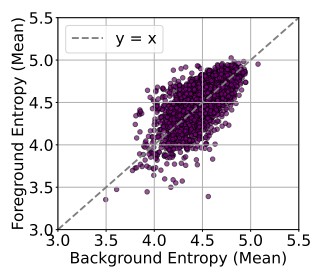

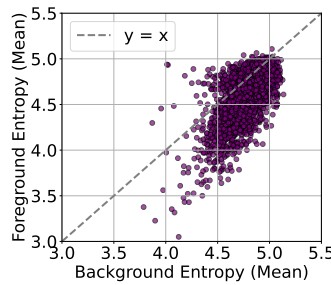

(a) ViT/B - Final Block Attention Entropies      (b) ALViT/B - Final Block Attention Entropies

Figure 3: Comparison of the image-level averaged foreground attention entropies vs background attention entropies of (a) MAE ViT/B baseline and (b) our ALViT/B model. Each point in the plots corresponds to an image on Imagenet-S-300 (Gao et al., 2022). ALViT/B has a higher attention entropy for the backgrounds for $83\%$ of the images, and ViT/B has a higher attention entropy for the backgrounds for only $43\%$, resulting in ViT/B missing critical, information-rich foreground signals.

Thus, these results quantify that the improvements of ALViT are not simply due to additional training capacity but a direct consequence of our design choices.

**Pretrained LLM Features are Essential for ALViT 's Gains.** Furthermore, to isolate the effects of architectural biases of appending an LLM block, we benchmarked another stronger baseline in Table 2, namely **ViT/B+Random LM1+LoRA (row e)**, identical to ALViT architecturally, only with a randomly-initialized and LoRA-adapted LLaMA 1 block. Echoing our observations from additional capacity ablations, ViT/B+Random LM1+LoRA only achieves on-par performance (row e: $83.25\%$ IN-1K) with ViT/B+MLP (LoRA Match), significantly falling behind ALViT.

**ALViT's Gains Are Robust with Different LLM Blocks and LLMs.** We showcase how ALViT's gains remain robustly high for a range of different LLM blocks in Table 3. Particularly in Table 3, we replace the default LLM block (LLaMA 1, $32^{nd}$) of ALViT with different blocks of LLaMA 1 and final blocks of different LLMs (Gemma 2 (Team et al., 2024), and LLaMA 3.1 (Grattafiori et al., 2024)). Along with a clear trend of improvement as the block index gets closer to the final block (rows a-d), the performance of ALViT remains largely invariant with the final blocks of different LLMs (rows e-g). These results establish that the architectural biases alone cannot account for the performance gains of ALViT, and the pretrained LLM representations are fundamental for the performance of ALViT.

## 5    ON THE BACKGROUND ROBUSTNESS OF ALViT

In this section, we establish an intriguing connection between the background robustness and the improved performance by our ALViT models, after analyzing the attention entropy patterns. Previously, Pang et al. (2023) hypothesized that frozen LLM block could be acting as a filter, amplifying the final contributions of the informative tokens as part of their *information filtering*

Table 4: Top-1 accuracy results of MAE pretrained models on Imagenet-9 adversarial backgrounds benchmark. The final three columns highlight the top-1 accuracy gap between different splits, a lower-better measure as denoted by the arrow ↓. **Bold** denotes best results.

| Model | Original | Same | Random | *Orig.-Same*↓ | *Orig.-Rand.*↓ | *Same-Rand.*↓ |
|---|---|---|---|---|---|---|
| MAE ViT/B | 96.5 | 87.8 | 83.2 | 8.7 | 13.3 | 4.6 |
| ALViT/B *(Ours)* | **96.6** | **89.2** | **85.3** | **7.4** | **11.3** | **3.9** |
| | +0.1 | +1.4 | +2.1 | −1.3 | −2.0 | −0.7 |

*hypothesis*. However, Pang et al. (2023) did not provide detailed discussions on the attention patterns, as they found the attention weights to be too noisy to provide insightful conclusions. We aim to bridge this gap and providing deeper insights on *how* ALViT performs the *information filtering*.

**ALViT Exhibits More Focused Attention Patterns.** We improve upon Pang et al. (2023)'s initial explorations and analyze the attention entropies of both the MAE ViT and our ALViT, thereby decrypting the previously under-explored attention patterns of ViTs utilizing LLM blocks. In particular, we quantify attention entropies through taking the post-softmax entropy of each row of the attention matrix, where each row corresponds to a spatial location on the feature map.Formally, denoting the input as $X \in \mathbb{R}^{T x d}$, and the query and key projection matrices as $W_Q \in \mathbb{R}^{d x d_k}, W_K \in \mathbb{R}^{d x d_k}$, the post-softmax attention matrix with its row-wise entropies are given by:

$$A = softmax\left[\frac{W_Q \cdot W_K^T}{\sqrt{d}}\right], \ \mathcal{H}(A_i) = -\sum_{j=1}^{T} A_{i,j} log(A_{i,j}). \tag{2}$$

We visualize the attention entropies on both image-level (Figure 2) and dataset-level on the Imagenet-S-300 dataset (Gao et al., 2022)(Figure 3). To achieve the dataset-level visualizations, we map the mask annotations of Imagenet-S-300 down to the resolution of our feature maps, and construct a binary mask to distinguish the foreground regions from the background regions. Then, we average the entropies of tokens belonging to the foreground vs background for each image.

In Figures 2 & 3, we observe a clear contrast between the attention entropies for the background and foreground regions for ALViT, with the majority of the samples having significantly higher attention entropy for the background regions. On the other hand, the average attention entropies are indifferent to background/foreground regions for the MAE ViT/B, highlighting its deficiency in differentiating informative regions from others. Finally, we present more visualizations in Section A.

**ALViT is More Robust Against Adversarial Backgrounds.** Inspired by these observations in the attention patterns, we benchmark our ALViT against the MAE ViT/B baseline on the challenging Imagenet-9, previously described in Section 4. Results in Table 4 show that the gains of ALViT significantly increase as the altered backgrounds become more challenging. In particular, for *Mixed Random* & *Mixed Same*, ALViT/B improves the performance by +**2.1** and +**1.4**. Finally, ALViT has significantly improved performance gaps between the original and background-altered splits, with gains reaching up to **2.0**.

## 6 CONCLUSION

In this work, we introduce Adapted-Language Vision Transformers (ALViT), a training framework that brings the semantic knowledge learned by text-only pre-trained LLM blocks into discriminative vision models. Our core contribution lies in a synergistic pre-training strategy: we leverage Masked Auto-Encoding (MAE) to learn rich visual representations from the ViT, while concurrently training Low-Rank Adaptation (LoRA) layers within an LLM block using the same MAE objective. This joint optimization process is crucial, guiding the ViT to produce LLM-friendly features while simultaneously enabling the LLM to effectively enhance these visual features with its vast semantic knowledge. Our comprehensive experiments demonstrate ALViT's efficacy. In image classification benchmarks, ALViT not only greatly pushes the frontier under its setting, but also shows a greatly improved robustness to domain shifts compared to strong baselines, including MAE ViTs. We show that while MAE pre-training provides a vital foundation, the LoRA-based adaptation of the LLM block, trained in tandem, is essential for unlocking performance gains. ALViT offers an effective and parameter-efficient pathway to harness the extensive knowledge of pre-trained LLMs for vision tasks.

## 7 REPRODUCIBILITY STATEMENT

Throughout our work, we utilized established libraries, frameworks and open-source repositories for our experiments. Furthermore, for all of the reproductions of our baselines, we directly utilized the code bases of the respective works, while adhering to all of their recommended training and evaluation settings, including the training configurations, hyperparameters and other associated details. All of these method-specific aspects, including the computational resources and dataset splits utilized in this work are clearly mentioned and discussed in Sections 4, D and E. Furthermore, for our primary ablations table (Table 2), we report the average performance scores and associated standard deviations for both our proposed ALViT and our baselines across 3 random seeds to ensure the robustness of our analyses and support reproducibility.

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

CONTENTS

## A   MORE VISUALIZATIONS WITH ATTENTION ENTROPIES

In Sections 4 and 5, we experimentally demonstrated the effectiveness of ALViT over the ViT baselines. Furthermore, we provided in-depth analysis regarding the background robustness properties of ALViT, where we demonstrated significant performance gains in Imagenet-9 (Xiao et al., 2020) with ALViT under adversarial backgrounds . Our empirical observations in Sections 4 and 5 were qualitatively grounded in the patterns we observe with the attention entropies of both ALViT and the ViT baseline. In particular, we showed that the foreground patches with ALViT exhibit significantly lower attention entropy compared to background patches, whereas the same distinction does not occur with the baseline ViT.

With the aim of solidifying these observations, we provide additional visualizations of the attention entropy patterns for both our ALViT and the baselines in this section. The visualizations and results presented in this section demonstrate that both the attention entropy patterns and the patch norms for ALViT provide significantly more salient visualizations compared to the ViT baseline (Section A.1), and that the observations made from the scatter plots in Section 5 generalize across all splits of the Imagenet-Segmentation benchmark (Section A.2).

### A.1   IMAGE-LEVEL ATTENTION ENTROPY VISUALIZATIONS

Here, we provide further details and image-level visualizations of attention entropy patterns along with the norms of the patches of both ALViT and our MAE pre-trained ViT baselines in Figure 4. Attention entropy patterns have been utilized in the context of neural network robustness in earlier works (Guo et al., 2023; Zhang et al., 2024). In these works, they provided litmus tests for measuring how focused the attention patterns of particular models are and how they relate to model robustness.

As stated in Section 5, we quantify the attention entropies through taking the post-softmax entropy of each row of the attention matrix, where each row corresponds to a spatial location, i.e., a patch, of the feature map, following the previous works using attention entropies (Zhai et al., 2023a).

Formally, denoting the input as $X \in \mathbb{R}^{T x d}$, and the query and key projection matrices as $W_Q \in \mathbb{R}^{d x d_k}, W_K \in \mathbb{R}^{d x d_k}$, the post-softmax attention matrix with its row-wise entropies are given by:

$$A = softmax\left[\frac{W_Q \cdot W_K^T}{\sqrt{d}}\right], \tag{3}$$

$$\mathcal{H}(A_i) = -\sum_{j=1}^{T} A_{i,j} log(A_{i,j}). \tag{4}$$

Figure 4: Visualized attention entropies across different blocks of both ALViT/B and the MAE pre-trained ViT/B baseline. ALViT/B simultaneously exhibits lower attention entropies and higher patch norms for foreground regions across all images compared to the ViT/B baseline, implying more focused attention patterns on these regions resulting in improved saliency in patch features. These results provide qualitative support to the background robustness behavior of ALViT/B over the ViT/B baseline. The brighter colors highlight patches with high attention entropy, whereas the darker colors highlight patches with low attention entropy for the **"Attention Entropies"** column and the brighter colors highlight the patches with higher norm whereas the darker colors highlight the patches with lower norm for the **"Patch Norms"** column.

Notably, we also average the attention entropies for each attention head, following the methodology of Zhai et al. (2023a). Finally, to further supplement our visualizations, we additionally extract the L2 norms of each patch and visualize it alongside the attention entropy patterns.

Following this quantification process, we visualize the attention entropyies along with the patch norms of both the final ViT block for both ALViT and the MAE pre-trained ViT baseline after finetuning on Imagenet-1K (Deng et al., 2009) in Figure 4. For Figure 4, we perform a per-image normalization for both the patch norms and attention entropies to achieve more interpretable visualizations. This corresponds to performing the normalizations based on the lowest and highest attention entropy score

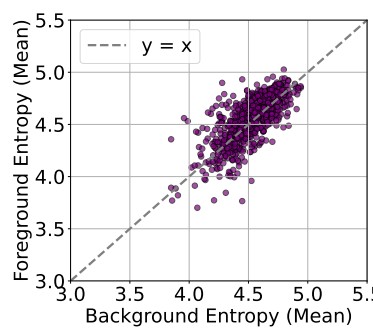 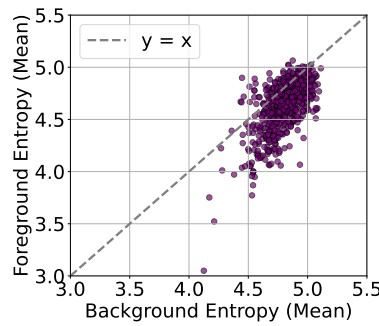

(a) ViT/B - Final Block Attention Entropies    (b) ALViT/B - Final Block Attention Entropies

Figure 5: Comparison of the image-level average foreground attention entropies vs the image-level average background attention entropies of (a) MAE ViT/B baseline and (b) our ALViT/B model. Each point in the plots corresponds to an image on Imagenet-S-50 dataset (Gao et al., 2022). For $84\%$ of the images, ALViT/B has a higher average attention entropy for the background regions, whereas this number drops to only $42\%$ for the ViT/B baseline.

or token norm value for each feature map separately, and follows the normalization strategy used for visualizations in Pang et al. (2023).

As it can be seen in Figure 4, ALViT exhibits much lower attention entropies for the patches belonging to foreground regions compared to ViT/B, providing further qualitative support for our observations in Section 5. Simultaneously, the patch norms are more salient and achieve better coverage of foreground regions for ALViT compared to ViT/B. This behavior is specifically important, since we are utilizing average pooling instead of relying on the [CLS] token, following the default implementation in the official MAE codebase. We refer the reader to Section D.3 for more details on this.

### A.2 ADDITIONAL ATTENTION ENTROPY SCATTER PLOTS

In Section 5, we presented the attention entropy scatter plots for the Imagenet-Segmentation-300 validation set (Gao et al., 2022). Here, we additionally present the scatter plots for the other two Imagenet-Segmentation variants (Gao et al., 2022), namely for Imagenet-Segmentation-50 validation set in Figure 5 and for Imagenet-Segmentation-919 validation set in Figure 6. Similar to the plots in Section 5, each point in Figures 5 and 6 correspond to the average attention entropy for each image where the y-axis highlights the average attention entropy for the foreground patches whereas the x-axis highlights the average attention entropy for the background patches.

These results closely mirror those in Section 5, where again a very clear distinction emerges between the average attention entropies for the background and foreground regions for our ALViT/B. On the other hand, the attention entropies are mostly the same for all regions of the MAE pretrained ViT/B baseline, regardless of whether they belong to a highly informative foreground region or not.

## B ADDITIONAL EXPERIMENTAL RESULTS

In this section, we present more experimental results on fine-grained visual recognition (Section B.1), additional ablations several of our design choices (Section B.2), LoRA-adapting the LLM under supervised-only training (Section B.3) and supplementary results on the Imagenet-Segmentation benchmark (Section B.4).

### B.1 FINE-GRAINED VISUAL RECOGNITION EXPERIMENTS

In Section 4, we demonstrated the effectiveness of ALViT on a variety of image classification benchmarks, under challenging scenarios. In this section, we further demonstrate the effectiveness of ALViT for fine-grained visual recognition tasks, namely on object detection and instance segmentation. The results on MS COCO, presented in Table 5, demonstrate ALViT's capability to enhance fine-grained visual recognition. Our ALViT/B model consistently outperforms the strong MAE ViT/B

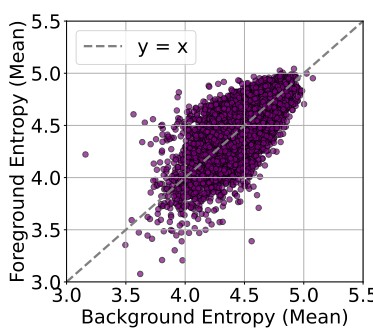 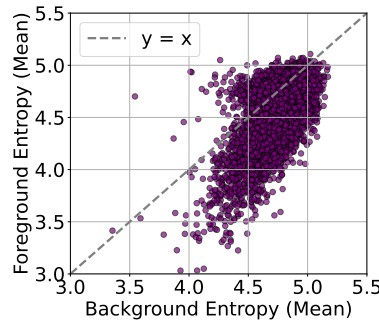

(a) ViT/B - Final Block Attention Entropies     (b) ALViT/B - Final Block Attention Entropies

Figure 6: Comparison of the image-level average foreground attention entropies vs the image-level average background attention entropies of (a) MAE ViT/B baseline and (b) our ALViT/B model. Each point in the plots corresponds to an image on Imagenet-S-919 dataset (Gao et al., 2022). For 83% of the images, ALViT/B has a higher average attention entropy for the background regions, whereas this number drops to only 44% for the ViT/B baseline.

Table 5: Object detection and instance segmentation results on MS COCO (Lin et al., 2014) dataset. Bounding box AP values are for the detection results whereas the mask AP values are for the instance segmentation results. **Bold** denotes the best result for each setting.

| Model | Bounding Box | | | Mask | | |
|---|---|---|---|---|---|---|
| | AP | $AP_{50}$ | $AP_{75}$ | AP | $AP_{50}$ | $AP_{75}$ |
| MAE ViT/B | 50.6 | 71.0 | 55.5 | 44.9 | 68.2 | 48.7 |
| ALViT/B *(Ours)* | **51.1** | **71.5** | **55.9** | **45.1** | **68.8** | **48.8** |
| | +0.5 | +0.5 | +0.4 | +0.2 | +0.6 | +0.1 |

baseline across all reported metrics for both object detection and instance segmentation. Specifically, ALViT/B achieves a bounding box AP of **51.1**, an improvement of +0.5 AP over the MAE ViT/B. For instance segmentation, ALViT/B achieves a mask AP of **45.1** (+0.2 AP improvement), with notable gains in $AP_{50}$ (+0.6).

## B.2 ADDITIONAL ABLATIONS ON DESIGN CHOICES

In this section, we provide supplementary ablations over several different design choices. Namely, Section B.2.1 shows that LoRA-adaptation can be more desirable to full finetuning of the LLM block under limited training set of Imagenet-1K, Section B.2.2 provides additional results using two LLaMA 1 blocks and finally Section B.2.3 discusses the results under different ranks of LoRA.

### B.2.1 ADAPTING VERSUS FULL FINETUNING THE LLM BLOCK

Following the success of LoRA-adapting the LLM block in ALViT, a natural question is what would the performance look like under full finetuning of the LLM block. To address this question, we trained a baseline on Imagenet-1K where we left the LLM block completely trainable during both the MAE pretraining and end-to-end finetuning stages. The results in Table 6 closely mirror the observations of Pang et al. (2023): The heavyweight LLM transformer block has a tendency to overfit, and regularizing it is very challenging, resulting in an even worse performance compared to the baseline with a completely frozen LLaMA 1 block.

### B.2.2 ALViT WITH MORE THAN ONE LLM BLOCK

Most of our explorations in Section 4 with ALViT was around using only a single LLM block. Another natural question is to ablate over the number of LLM blocks to be included to observe the possibility of reaping the benefits of the additional pretrained capacity. To investigate this aspect, we

Table 6: Full finetuning of the LLM block can be undesirable under limited training sets, even though it has a much greater number of trainable parameters. "ViT/B + Trainable LM1" denotes the baseline with a fully-finetuned LLaMA 1 block. All results indicate Imagenet-1K top-1 accuracy. **Bold** denotes the better result.

| Model | Trainable Params. | Accuracy |
|---|---|---|
| ViT/B | 86.8M | 83.2 |
| ViT/B+Frozen LM1 | 92.9M | 83.1 |
| ViT/B+Trainable LM1 | 295.2M | 82.2 |
| ALViT (*Ours*) | 93.1M | **83.6** (+0.4) |

Table 7: Multiple LLM blocks could be promising, though they likely require a more careful hyperparameter selection. "ALViT 2-LM1" denotes the baseline with two LoRA-adapted LLM blocks. All results indicate Imagenet-1K top-1 accuracy. **Bold** denotes the better result.

| Model | LLaMA 1 Blocks | Accuracy |
|---|---|---|
| ALViT 2-LM1 | 31 & 32 | 83.5 |
| ALViT (*default*) | 32 | **83.6** |

trained another ALViT variant with the final two LLaMA 1 blocks instead of only the final block. We chose the final two blocks of LLaMA 1 following the results in Table 3, and report the results in Table 7.

The results in Table 7 highlight that although having multiple LLM blocks is still significantly better than our previous baselines, it does not provide immediate performance improvements over our default configuration of using a single LLM block. Following our quantitative observations of the two-blocks baseline simultaneously having higher training and testing errors, we hypothesize that its performance could be further improved with a more careful hyperparameter search, though we could not perform these experiments due to computational constraints.

### B.2.3 Ablations on LoRA Hyperparameters

In this section, we present additional results with different rank and $\alpha$ hyperparameters for the LoRA layers of ALViT. The results in Table 8 show that setting $r = \alpha = 16$ is sufficient for achieving the performance gains, since the performance seems to saturate beyond this rank, as evidenced by the similar performances of the $r = \alpha = 16$ and $r = \alpha = 32$ configurations. Furthermore, the $r = \alpha = 8$ seems to be insufficient for fully reaping the benefits of the pretrained LLM representations. Accordingly, we opted for $r = \alpha = 16$ for our experiments in Section 4.

### B.3 Adapting the LLM Under Supervised-only Training

In Section 4, we demonstrated the effectiveness of employing Masked Auto-Encoding (MAE) to pretrain the ViT for richer visual representations, while concurrently training Low-Rank Adaptation (LoRA) layers within the LLM block using the same MAE objective.

Following our results and ablations in Section 5, a natural question may arise regarding how well a concurrent training strategy of Low-Rank Adaptation (LoRA) layers, and the ViT could work *without* the critical MAE pretraining phase. In this section, we investigate this question by including Low-Rank Adaptation (LoRA) layers on top of the architecture proposed in Pang et al. (2023) in a supervised-only setting.

The results of both our reproduction of Pang et al. (2023)'s model and the LoRA-adapted version of it are presented in Table 9. For Table 9, we train both Pang et al. (2023)'s LM1+ViT/B and its LoRA-adapted version in a supervised-only setting on Imagenet-1K with three random seeds while adhering to all training settings in Pang et al. (2023). Furthermore, we directly utilize their code-base

Table 8: Experiments with different LoRA hyperparameters highlight that rank and $\alpha$ $r = \alpha = 16$ is sufficient enough for effectively adapting the LLM block. All results indicate Imagenet-1K top-1 accuracy. **Bold** denotes the best result.

| Model | LoRA Rank | Accuracy |
|---|---|---|
| | $r = \alpha = 8$ | 83.3 |
| ALViT | $r = \alpha = 16$ (*default*) | **83.6** |
| | $r = \alpha = 32$ | **83.6** |

Table 9: Adapting the LLM block in the ViT of Pang et al. (2023) with a supervised-only training regime. Each reported value is an average runs with three random seeds and the subscript $\pm$ denotes the standard error for each setting. Note that we report two significant digits in the decimal for highlighting the effect of standard errors in contrast with other tables. **Bold** denotes the best result.

| Model | Average Accuracy |
|---|---|
| LM1+ViT/B | $80.51_{\pm 0.07}$ |
| LoRA LM1+ViT/B | $80.63_{\pm 0.09}$ |

[1], and merely inject trainable LoRA layers to its LLM block, similar to the methodology of ALViT presented in Sections 3 and 4. Finally, we report the average accuracy across the seeds with the accompanying standard error values, i.e. the standard deviation of the accuracy values divided by the number of different seeds.

From Table 9 we observe that the LoRA version achieves a slightly improved performance compared to the LM1+ViT/B. Concretely, the LoRA-adapted version of Pang et al. [2023]'s LM1+ViT/B has a +0.12 better accuracy compared to the completely frozen LM1+ViT/B. These results highlight that LoRA could potentially address the training instabilities that arise when directly finetuning the LLM block even under a supervised-only setting, though the performance improvements are much less pronounced compared to ALViT 's improvements over the MAE ViT/B baselines.

### B.4   ADDITIONAL RESULTS ON IMAGENET-SEGMENTATION BENCHMARK

In this section, following up from the qualitative observations with respect to token norms in Figure 4, we provide additional quantitative evidence that ALViT forms more salient patch features compared to the MAE pretrained ViT baseline. In particular, we compare the binary mask IoUs of the patch features with respect to the ground truth segmentation masks presented in all three of the Imagenet-Segmentation splits (Gao et al., 2022) in Table 10, following the methodology of Pang et al. (2023).

In Pang et al. (2023), the authors leveraged a method to extract **magnitude** and **frequency** components from token features to generate pseudo-masks. Specifically, the magnitude component is obtained by taking the L2 norm of each token feature vector after centering, while the frequency component is obtained by taking the norm of the difference between the angle of a token feature vector and the average angle across all tokens in the same input, following a Fast Fourier Transform (FFT) (Pang et al., 2023).

Once these components are extracted for each token, binary pseudo-masks are created by applying a fixed threshold to either the magnitude or frequency values, assigning a binary label to each patch accordingly. This fixed threshold is determined empirically and individually for each component of each model, following the approach of Pang et al. (2023). Finally, Pang et al. (2023) downsample the ground-truth segmentation masks in the ImageNet-Segmentation-50 dataset (Gao et al., 2022) to match the resolution of the model's feature map. A cell in the downsampled mask is assigned a value of **1** if it overlaps with the original high-resolution mask. Concretely, this resolution is $14^2$ for a ViT/B with patch size set to 16, with $224^2$ resolution for images of Imagenet (Deng et al., 2009).

---

[1]https://github.com/ziqipang/LM4VisualEncoding

Table 10: Mask IoUs of the final ViT block for each model with respect to the Imagenet-Segmentation (Gao et al., 2022) segmentation annotations. Frequency column shows the results when the frequency component of the token features are used for obtaining the binary masks whereas the magnitude column shows the results when the magnitude component is used. **Bold** denotes the best result.

| Model | INS-50 | | INS-300 | | INS-919 | |
|---|---|---|---|---|---|---|
| | Frequency | Magnitude | Frequency | Magnitude | Frequency | Magnitude |
| MAE ViT/B | 39.8 | 42.3 | 40.9 | 42.8 | 40.8 | 42.8 |
| ALViT/B *(Ours)* | **41.3** | **43.8** | **42.3** | **43.5** | **42.1** | **43.3** |
| | +1.5 | +1.5 | +1.4 | +0.7 | +1.3 | +0.5 |

For more details regarding the frequency and magnitude components or the mask IoU measures, we refer the reader to the Appendix A.3 and Appendix A.5 of Pang et al. (2023).

As evidenced in Table 10, ALViT has a higher IoU not only across all three subsets of the Imagenet-Segmentation benchmark, but also with *both* of the frequency and magnitude components. Notably, the average improvement in terms of IoU gains for the frequency component of ALViT compared to the ViT/B is $+1.4$, while for the magnitude component of ALViT compared to the ViT/B is $+0.9$. These results further solidify the quantitative and qualitative analyses provided in Section 5 of the main work, while also grounding our qualitative observations in Section A.

## C    FURTHER DISCUSSIONS ON RELATED WORKS

In this section, we discuss the closely related works to our work in more detail while highlighting the key differences, similarities and orthonogal directions between them and our ALViT framework.

### C.1    INFORMATION FILTERING HYPOTHESIS

An important contribution of Pang et al. (2023) was the introduction of the *information filtering hypothesis*. Information filtering hypothesis was proposed as a potential explanation towards how a frozen LLM block could enhance the visual features for visual recognition tasks. Particularly, Pang et al. (2023) first follows from the DeiT (Touvron et al., 2021) family of models and perform classification based on the [CLS] token. Then, the authors made the claim that to achieve a better performance compared to the vanilla ViT/B, either the attention weights should be improving or the informative tokens should be getting amplified by the LLM block.

Formally, denoting the set of visual tokens with $v \in V$, attention weights of the final ViT block with $w_v$, the processed visual token $v$ by the first linear layer following the ViT block as $M_L^1(z[v]) = z_v^1[v]$ and the the processed [CLS] token following the LLM block with $z'_{[CLS]}$, the hypothesis proposes the following correlation:

$$z'_{[CLS]} \propto \sum_{v \in V} w_v(M_L^2 \cdot M_{LLM} \cdot z_v^1[v]), \tag{5}$$

with the assumption that the $M_L^2 \cdot M_{LLM} \cdot M_L^1$ is a linear projection.

However, Pang et al. (2023) made the qualitative observation that the attention weights, $w_v$, were noisy, thus concluding that the $M_L^2 \cdot M_{LLM}$ projection must be amplifying the most informative tokens.

While ALViT differs from Pang et al. (2023)'s frozen-LLM-appended ViTs in several key architectural and training-related details, our work also leverages the pretrained LLM representations to improve discriminative visual recognition asks. In addition, as we have also discussed in Section 5 and Section A, ALViT exhibits strong robustness against adversarial backgrounds compared to the baselines, an potential consequence of the information filtering hypothesis. Coupled with our attention entropy observations, analyses we present in Sections 5 and A can be thought in a similar spirit with the information filtering hypothesis where we provide complementary discussions.

## C.2 PRETRAINED LLM LAYERS AND GRADIENT COHERENCE IN VISION TRANSFORMERS

Another recent work investigating the underlying mechanisms behind how a frozen LLM block improves the visual recognition performance is proposed by Bai et al. (2025), where the authors approach from a gradient dynamics perspective.

In particular, Bai et al. (2025) broadly borrowed the architecture of Pang et al. (2023) and demonstrated that the gradient flow from different samples towards the weights of the model are more aligned in the presence of the frozen LLM block. The authors quantified this alignment through demonstrating improved gradient-signal-to-noise ratio (GSNR) under the presence of the LLM block. Notably, GSNR for a given parameter is the ratio between the squared expected value and the variance of the its gradient. A high GSNR is also tied with improved generalization for machine learning models (Liu et al., 2020; Michalkiewicz et al., 2023), and thus is a desirable property.

Bai et al. (2025) also showed that this effect is more pronounced towards layers closer to the LLM block, and that the similar representations between the ViT blocks and the LLM block could be indicative of improvements. Following up from this observation and taking inspirations from Tiwari & Shenoy (2023), Bai et al. (2025) then proposes an auxiliary training objective with the aim of removing the additional inference costs incurred by the LLM block. This auxiliary training objective distills the representations of the frozen-LLM-appended ViT to a vanilla ViT through a similarity loss in-between (Hinton et al., 2015).

Bai et al. (2025)'s work thus presents an orthogonal direction, and a potentially interesting future work for our work. Particularly, their auxiliary loss could be combined with our ALViT as the teacher model for distilling the vanilla ViT, as ALViT has stronger visual recognition performance compared to the baseline teacher models utilized in Bai et al. (2025).

## C.3 COMPARISONS WITH LARGE VISION-LANGUAGE MODELS

**Large language models for visual tasks.** Large language models (LLMs) are utilized in unison with visual encoders in numerous different multi-modal architecture settings. The most common branch of these works involve using the LLMs as the *textual decoders* (Li et al., 2022a; 2023; Liu et al., 2023; Chen et al., 2024b;c; Alayrac et al., 2022), where they are preceded by visual encoders. In these works, encoder-processed visual tokens are simply projected to the text decoder (Li et al., 2022a; Liu et al., 2023) or fused through additional cross-modal layers (Alayrac et al., 2022).

All of the aforementioned works demonstrate that LLMs can process vision-originating data, given that they are processed by a separate visual encoder (Liu et al., 2023; Li et al., 2022a) or trained jointly from scratch on vast amounts of data in multiple stages (Diao et al., 2024; Wang et al., 2025; Luo et al., 2024). Our work is inspired by the success of the aforementioned approaches, while differentiating in several key aspects. Namely, our goal is to effectively leverage LLM transformer blocks and Self-Supervised Learning (SSL) for improving the performance of vision transformers (Dosovitskiy et al., 2020), without relying on language-aligned visual encoders (e.g. CLIP (Radford et al., 2021)) or requiring language inputs.

**Large monolithic vision language models.** A novel branch of works which are architecturally related to our work are foundation vision-language models aiming to contain both the vision and language modalities inside of a large monolithic transformer (Diao et al., 2024; 2025; Wang et al., 2025; Luo et al., 2024; Bavishi et al., 2023; Chen et al., 2024a). These works are differ from other *encoder-decoder* (Li et al., 2022a; Liu et al., 2023; Li et al., 2023; Yu et al., 2022; Wan et al., 2024) or *two-tower encoder* (Radford et al., 2021; Tschannen et al., 2025; Zhai et al., 2023b) alternatives, where they enforce varying degrees of intra-block parameter sharing between the transformers for each modality. To exemplify, while Fuyu (Chen et al., 2024a), EVEv1 (Diao et al., 2024) all share the majority of the Transformer components, EVEv2 (Diao et al., 2025) only shares the self-attention block while having modality-specific layer norm (LN) (Ba et al., 2016) and MLP blocks inside each transformer.

While the monolithic vision-language models share some similarities with our work, they also differ in several key aspects. Namely, while our goal is to achieve stronger *discriminative* visual performance, these works mainly target generative domains, such as as visual question answering (VQA) (Goyal et al., 2017b) or image captioning (Chen et al., 2015).

In addition, all of the monolithic vision-language model works (Diao et al., 2024; 2025; Wang et al., 2025; Luo et al., 2024; Bavishi et al., 2023; Chen et al., 2024a) involve jointly training both the language and vision-related components on vast amounts of multi-modal data in multiple training stages with multiple objectives. In our work we merely adapt our LLM block with simple and cost-effective LoRA layers with a unified MAE objective without requiring any language-specific inputs or additional objectives, thereby achieving strong unimodal performance without extensive multimodal training.

# D    TRAINING DETAILS OF EXPERIMENTS

In this section, we describe the architectural details, hyperparameter settings and other training details that we adhered to throughout this work.

## D.1    ARCHITECTURAL DETAILS

Throughout our experiments, we utilize the ViT/B as our encoder from Dosovitskiy et al. (2020) with a patch size of 16x16, which consists of 12 Transformer (Vaswani et al., 2017) blocks and has a hidden size of 768. In addition, for ALViT, we primarily utilize the $32^{nd}$ (i.e the final) Transformer (Vaswani et al., 2017) block of the smallest LLaMA 1 (Touvron et al., 2023a) model with 7 billion parameters and a hidden size of 4096, unless otherwise stated for ablations. We choose this block of LLaMA 1 following its success in similar works (Lai et al., 2024; Pang et al., 2023; Bai et al., 2025) and our empirical observations following our ablations in Section 4.2. There are also two additional linear projections *without* any non-linearities or additional activations around the LLaMA 1 block to allow matching the hidden dimensions of the ViT and the LLaMA 1.

During the pretraining stage, for both ALViT and our baselines, we additionally employ a lightweight Transformer (Vaswani et al., 2017) decoder, which consists of 8 blocks and has a hidden size of 512. The design of both the ViT/B encoder and the lightweight decoder closely mirror the original MAE design with no changes with the exception of the LLaMA 1 block and the linear projections around it.

For LoRA-related hyperparameters, we performed our experiments with a rank of $r = 16$ throughout our work, following the common usage of this parameter in the literature (Hu et al., 2022). Similarly, following Hu et al. (2022), we set the alpha parameter the same as our rank parameter, *i.e.* $r = \alpha = 16$. We also provide additional ablations on the rank hyperparameter and $\alpha$ hyperparameters in Section B.2. Finally, we did not adjust a particular learning rate scheduling mechanism for LoRA layers, and they directly followed the learning rate schedule of the final ViT block.

## D.2    SELF-SUPERVISED PRETRAINING

For all of our self-supervised pretraining experiments, we directly adhere to all of the settings presented in the original MAE work (He et al., 2022), while training both our models and the baselines for 800 epochs.

Namely, this corresponds to having a batch size of 4096, base learning rate of $1.5e\text{-}04$, with cosine annealing scheduling (Loshchilov & Hutter, 2016). In addition, we used the AdamW optimizer (Kingma, 2014; Loshchilov & Hutter, 2017) with $\beta_1 = 0.90$ and $\beta_2 = 0.95$ (Chen et al., 2020a), coupled with 40 warm-up epochs (Goyal et al., 2017a) and a weight decay of 0.05. Finally, we also apply a random resized crop augmentation, utilized a random masking ratio of 75% for masking the encoder inputs, and a normalized pixel version of mean squared error (MSE) between the reconstructed and the ground truth images as the objective.

## D.3    END-TO-END FINETUNING FOR CLASSIFICATION

Analogously with Section D.2, we directly adhere to all of the settings presented in the original MAE work (He et al., 2022). Namely, this corresponds to having a batch size of 1024, learning rate of $1.e\text{-}03$, with cosine annealing scheduling (Loshchilov & Hutter, 2016). In addition, we used the AdamW optimizer (Kingma, 2014; Loshchilov & Hutter, 2017) with $\beta_1 = 0.90$ and $\beta_2 = 0.999$ (Chen et al., 2020a), coupled with 5 warm-up epochs (Goyal et al., 2017a) and a weight decay of 0.05. Differing from the pretraining stage, here we have a layer-wise learning rate decay value of

0.75 (Bao et al., 2021; Clark et al., 2020), label smoothing of 0.1 (Szegedy et al., 2016) and a drop path rate of 0.1 (Huang et al., 2016). Finally, we also applied *mixup* (Zhang et al., 2017) with 0.8, *cutmix* Yun et al. (2019) with 1.0, and Randaugment with (9, 0.5) (Cubuk et al., 2020).

Notably, we utilize average pooling setting instead of relying on the [CLS] token for performing classification. We do so, following the official MAE Github repository's [2] report of potential instabilities in the loss values [3] when the [CLS] token was used with Pytorch (Paszke, 2019).

### D.4 TRAINING FOR FINE-GRAINED VISUAL RECOGNITION

Our fine-grained visual recognition experiments mostly follow from the ViTDet framework (Li et al., 2022b), which is a competitive fine-grained visual recognition framework achieving competitive results with plain ViT backbones (Dosovitskiy et al., 2020) with respect to previously-stronger hierarchical counterparts, such as the Swin Transformer (Liu et al., 2021). ViTDet framework involves taking an MAE pretrained plain ViT backbone, a following simple feature pyramid structure Lin et al. (2017) and a Mask R-CNN (He et al., 2017) as the final detection/segmentation head. Notably, achieving competitive fine-grained visual recognition results is very hard with supervised-only ViT backbones, with neither of Imagenet-1K nor Imagenet-22K supervised-pretrained ViT/B models achieving better results than a randomly initialized ViT/B, further highlighting the necessity of self-supervised pretraining for achieving strong fine-grained visual recognition.

Finally, the entire model, with the notable exception of the LLM block that we always keep frozen and merely adapt through the LoRA layers, including the ViT/ALViT backbones, is trained jointly on the COCO training set (Lin et al., 2014), with a batch size of 64, a learning rate of $1.5e$-04, weight decay of 0.1, drop path rate of 0.1 and for 100 epochs. For both our baselines and ALViT, we directly adhere to the settings of Li et al. (2022b), and do not change any hyperparameters. We implemented our ALViT/B ViTDet and benchmarked both ALViT and our baselines on the mmdetection library (Chen et al., 2019).

### D.5 COMPUTATIONAL RESOURCES

For all of the aforementioned experiments, we ran our experiments on 32 NVIDIA A100 GPUs. For MAE pretraining described in Section D.2, both the ALViT and the baseline experiments take approximately 30 hours. For both the end-to-end finetuning for classification and the fine-grained visual recognition training experiments, both ALViT and the baseline experiments take approximately 24 hours.

## E  DETAILS OF THE USED DATASETS

In this section, we provide the details of the datasets we used for our experiments and other quantitative analyses, while clarifying the exact splits and settings we report our results on.

**Imagenet-1K.** Imagenet-1K (Deng et al., 2009) consists of $1.2M$ training and $50K$ validation images, belonging to 1000 different classes. Following the conventional approach (He et al., 2016; Dosovitskiy et al., 2020), we used the resized ($224^2$) images for both training and evaluation. We performed the MAE pretraining exclusively on Imagenet-1K training set, for all of our classification and fine-grained visual recognition experiments, following our baselines (He et al., 2022; Li et al., 2022b).

**Imagenet-9.** Imagenet-9 (Xiao et al., 2020) consists of images of the 9 super-classes from the original Imagenet-1K validation set (Deng et al., 2009), and aims to measure the background over-reliance of deep learning models in an evaluation-only setting. In particular, Imagenet-9 has contains numerous splits, such as the *original*, *mixed random*, *mixed same*, and *mixed next*. The first of these splits, *original* consists of the unaltered images belonging to the 9 super-classes, with their original backgrounds. On the other hand, *mixed random, mixed same*, and *mixed next* consist of images with altered backgrounds. For *mixed random*, the background of each image is replaced with the

---

[2]https://github.com/facebookresearch/mae/tree/main
[3]https://github.com/facebookresearch/mae/blob/main/FINETUNE.md

background of another image from a random super-class, for *mixed next*, the background of each image is replaced with the background of another image from the next super-class ordered with respect to their numerical IDs, and for *mixed same* the background of each image is replaced with the background of another image from the same super-class.

In Imagenet-9 (Xiao et al., 2020), while it is desirable to obtain high performance on the clean *original* set, it is crucial to obtain high performance on the splits with altered backgrounds for demonstrating the robustness of the models, thereby achieving a smaller *background accuracy gap*. Finally, for our evaluations, we utilized the *original* split for benchmarking the clean accuracy of the models in our work, while comparing it to the accuracies in *mixed random* and *mixed same* splits for measuring the background over-reliance of models.

**Imagenet-Segmentation.**   Imagenet-Segmentation (Gao et al., 2022) consists of the images and associated high-quality segmentation masks of the original Imagenet-1K (Deng et al., 2009) images. It has 3 splits of different sizes, Imagenet-Segmentation-50 as the 50 class subset with 752 validation images, Imagenet-Segmentation-300 as the 300 class subset with $4K$ validation images, and Imagenet-Segmentation-919 as the 919 class subset with $12K$ validation images. Notably, the largest 919 split does not contain the images of non-segmentable 81 classes from the original Imagenet-1K splits (Gao et al., 2022). We re-purpose this dataset in the same format as Pang et al. (2023), though including additional results and visualizations on the more challenging Imagenet-Segmentation-300 and Imagenet-Segmentation-919 instead of limiting the analyses to the limited Imagenet-Segmentation-50 split as in Pang et al. (2023).

**Imagenet-C.**   Imagenet-C (Hendrycks & Dietterich, 2019) benchmark is an evaluation-only benchmark consists of synthetically corrupted images belonging to the Imagenet-1K validation (Deng et al., 2009) set. In particular, there are 15 benchmark corruptions, namely 4 noise corruptions (*gaussian noise, shot noise, impulse noise*), 4 weather-related corruptions (*snow, frost, fog, brightness*), 4 blurring corruptions (*defocus, glass, motion, zoom*) and 3 digital corruptions (*contrast, elastic transform, pixelate*). Furthermore, there are 4 additional corruptions, namely *gaussian blur, spatter, saturate, speckle noise*, bringing the total to 19. For each of these corruptions, there are 5 severity levels, with higher number indicating tougher corruptions. In our experiments, we report the average results on all of the aforementioned corruptions with all of their severities for a more comprehensive evaluation.

**Imagenet-A.**   Imagenet-A (Hendrycks et al., 2021b) is an adversarially-designed benchmark consisting of images from Imagenet-1K validation set, where the majority of the Imagenet-1K-trained classifiers fail. Notably, it has 200 super-classes instead of the full 1000 classes of the Imagenet-1K benchmark, where the super-classes were explicitly constructed in a way that confusing them would be beyond a simple confusion of similar classes.

**Imagenet-SK.**   Imagenet-SK (Wang et al., 2019) consists of $50K$ images of sketches of Imagenet-1K classes, 50 for each of the 1000 classes of the Imagenet-1K validation set. Notably, images of Imagenet-SK are *black and white* sketches, posing a challenge due to their lack of texture and color information.

**Imagenet-V2.**   Imagenet-V2 (Recht et al., 2019) is a benchmark proposed to measure the broader generalization capabilities of Imagenet-1K-trained models. It also has samples for the same 1000 classes of the Imagenet-1K, though with specifically curated examples where the majority of the Imagenet-1K-trained classifiers tend to fail. Among its different variants, we utilized the *matched frequency* version, as it is proposed to be the default setting in Recht et al. (2019).

**Imagenet-R.**   Imagenet-R (Hendrycks et al., 2021a) is a $30K$ image domain-generalization benchmark for Imagenet-1K-trained classifiers. It contains *"renditions"* of images belonging to Imagenet-1K classes, in the form of images of sculptures or paintings, with drastically different textures, and other often-helpful image-level statistics.

**MS COCO.**   MS COCO (Lin et al., 2014) is an object detection and instance segmentation benchmark for benchmarking fine-grained visual recognition capabilities of deep learning models. Among its variants, we train both ALViT/B with ViTDet (Li et al., 2022b) and ViT/B with ViTDet (Li et al.,

2022b) models on COCO2017 training set and report our results on the COCO2017 validation set, following the common practice (Li et al., 2022b; Carion et al., 2020; He et al., 2017).

# F    LIMITATIONS

Even though ALViT benefits from the combined powers of self-supervised learning with MAE and the LoRA-adapted pretrained LLM representations for discriminative computer vision tasks, it also inherits the drawbacks of these works. First, while ALViT does not introduce a significant training overhead over the ViT baselines, the computational costs of MAE pretraining is still substantial, even though it is drastically cheaper compared to alternative self-supervised learning methods (Oquab et al., 2023; Caron et al., 2021). In addition, the two-stage training nature of our framework can be undesirable for the practitioners of downstream applications. However, the addition of the LLM block inevitably introduces an increase in inference time over our vanilla ViT baselines, which may limit its usage on downstream tasks requiring real-time processing. Regardless, as highlighted by ALViT's significant improvements over models of similar sizes (*i.e.* the randomly-initialized LLM block in Section 4 and the trainable LLM block ablations of B.2.1), ALViT's combination of pretrained LLM representations, the MAE objective and the LoRA adaptation could be highly desirable if the real-time inference costs are not of major concern.

# G    LLM USAGE STATEMENT

LLM assistance was only consulted for basic aiding and polishing of the writing, such as finding typos, spell checking, or basic word choices no more than the order of a few words to convey the message of the work more clearly. No LLM usage beyond these purposes was utilized.

