# OpenReview forum: "Adapted-Language ViT: Empowering Self-Supervised Vision Transformers with LLMs"
_ICLR.cc/2026/Conference — Submitted to ICLR 2026_

### Official Review · Reviewer_T4ti · 2025-10-21

**Soundness:** 2
**Presentation:** 3
**Contribution:** 2
**Rating:** 4
**Confidence:** 4

**Summary:**

This paper introduces AL-ViT (Adapted Language Vision Transformer), a framework designed to improve multimodal alignment between visual and linguistic representations in large-scale Vision Transformers. The key idea is to co-adapt a ViT and an LLM block through joint MAE-based self-supervision and LoRA-based LLM adaptation, achieving the alignment between modalities.

**Strengths:**

**Simple Solution:** The training pipeline (MAE pre-training + LoRA adaptation) is well-structured and straightforward.

**Good presentation:** The paper is well-organized and easy to follow.

**Weaknesses:**

**Incremental Technical Novelty:** The method primarily combines existing techniques — MAE for visual pretraining, LoRA for efficient adaptation, and LLM fusion from LM4Vision — into a joint optimization scheme. There is no fundamentally new algorithmic or theoretical contribution beyond this integration.

**Unclear Mechanistic Insight:** The paper hypothesizes that joint MAE-LoRA training bridges modality mismatch, but provides no formal analysis or theoretical grounding. The attention-entropy study is qualitative and does not establish causality.

**Cost and Gain:** The addition of an LLM block (even partially adapted) introduces large computational overhead and complexity for minimal gain. There is no clear analysis of training cost, FLOPs, or memory increase.

**Weak Motivation for LLM Use:** Since the LLM is frozen and only LoRA-adapted via reconstruction loss (not text supervision), it is unclear whether the language knowledge actually benefits the visual features — especially without textual context or captions.

**Questions:**

Please refer to the weakness section.

---

> ### Author Response · Authors · 2025-11-21
> **Response for W1 and W2 (1/2)**
>
> We thank the reviewer for finding our work to have a **well-structured training pipeline with well-organized and easy to follow narrative.**
>
> Below, we present our responses to each of the issues raised by the reviewer.
>
> ---
>
> **W1: Incremental novelty**
>
> As acknowledged by **reviewers Ajvx and 4iid,** our primary contribution is that we propose an effective visual backbone by successfully combining the powerful self-supervised MAE pretraining with the frozen knowledge of an LLM. We show that our fusion leads to robust gains on numerous challenging benchmarks (up to **\+0.5** on Imagenet-1K, **\+2.2** on Imagenet-A and up to **\+2.1** on Imagenet-9). Unlike LM4Vision \[A\], we show that simply inserting an LLM block does not help a strong MAE-pretrained backbone; the remarkable gains appear only when the block is equipped with lightweight LoRA adaptations. We also strengthen our experimental results with intriguing analyses on *why* our method works by analyzing *how* the ViT leverages the inherited LLM representations.
>
> Note that MAE is a very strong self-supervised pretraining strategy greatly improving over many self-supervised (e.g. BeiT \[B\]) and supervised baselines especially for dense prediction tasks like object detection \[C\]. Importantly, supervised-only pretrained backbones often result in sub-optimal performance, even against randomly-initialized visual backbones \[C\].
>
> Finally, we would like to note that ALViT is a more performant ViT backbone that could provide an improved starting point for more recent state-of-the-art masked-image modeling based approaches for computer vision, such as EVA-variants \[D, E\]. Therefore, we believe that ALViT is an interesting, novel idea for attaining a better alternative to vanilla ViT.
>
> ---
>
> **W2: Unclear mechanistic insight**
>
> As acknowledged by **reviewers BRBi, Ajvx and 4iid,** through our attention entropy analyses, we explain the driving factors behind the performance gains of ALViT, where we highlight that patches under ALViT demonstrate superior ability to distinguish foreground and background regions. Notably, in addition to the qualitative visualizations and providing dataset-level statistics, **we quantitatively show that ALViT achieves up to \+2.1 gains over MAE ViT/B on Imagenet-9 adversarial backgrounds benchmark**, a benchmark explicitly designed to analyze the foreground/background reliance of different models.
>
> ---
>
> \---
>
> \[A\] Pang, Z., Xie, Z., Man, Y., & Wang, Y. X. (2023). Frozen transformers in language models are effective visual encoder layers. arXiv preprint arXiv:2310.12973.
>
> \[B\] Wang, Wenhui, et al. "Image as a foreign language: Beit pretraining for vision and vision-language tasks." Proceedings of the IEEE/CVF Conference on Computer Vision and Pattern Recognition. 2023\.
>
> \[C\] Li, Y., Mao, H., Girshick, R., & He, K. (2022, October). Exploring plain vision transformer backbones for object detection. In European conference on computer vision (pp. 280-296). Cham: Springer Nature Switzerland.
>
> \[D\] Fang, Y., Wang, W., Xie, B., Sun, Q., Wu, L., Wang, X., ... & Cao, Y. (2023). Eva: Exploring the limits of masked visual representation learning at scale. In Proceedings of the IEEE/CVF conference on computer vision and pattern recognition (pp. 19358-19369).
>
> \[E\] Fang, Y., Sun, Q., Wang, X., Huang, T., Wang, X., & Cao, Y. (2024). Eva-02: A visual representation for neon genesis. Image and Vision Computing, 149, 105171\.

---

> ### Author Response · Authors · 2025-11-21
> **Response for W3 and W4 (2/2)**
>
> **W3: Cost and gain**
>
> The performance improvements brought in by ALViT are statistically significant and are not trivial, as demonstrated by the standard deviation of the performance across numerous Imagenet benchmarks in Table 1\. Throughout our experiments with Imagenet-1K, Imagenet robustness, Imagenet-Segmentation and MS COCO benchmarks, ALViT has shown consistent and steady improvement over all of our baselines across different vision tasks, achieving up to **\+0.5** on Imagenet-1K, **\+2.2** on Imagenet-A, **\+2.1** on Imagenet-9, up to **\+1.5** on Imagenet-Segmentation and **\+0.5** on MS COCO object detection task.
>
> To further solidify our results and demonstrate the strengths of ALViT, we also performed experiments with MILAN \[F\], a stronger masked-image modeling approach which aims to reconstruct the CLIP visual encoder features instead of original image features as in MAE. The results we obtained show that ALViT achieves **\+7.0 improvement on kNN accuracy (81.0% vs 74.0%) and \+3.5 improvement on linear probing accuracy (82.2% vs 78.7%)** over the vanilla MILAN ViT.
>
> Accordingly, we believe that our method has significant potential for future applications under an even wider range of applications.
>
> Furthermore, below we detail the total number of parameters for both ALViT and the baselines we considered, where we show that ALViT achieves a more desirable trade-off with a comparable number of trainable parameters:
>
> **Imagenet-1K Mean Top-1 Accuracy with Number of Total and Trainable Parameters**
>
> | Model | Accuracy | Total Parameters | Trainable Parameters |
> | :---- | :---- | :---- | :---- |
> | ViT/B | 83.11% | 86.6M | 86.6M |
> | ViT/B \+ Random LM1+LoRA | 83.25% | 295.2M | 93.2M |
> | ALViT/B | **83.63%** | 295.5M | 93.2M |
>
> Finally, to provide a more transparent discussion around ALViT’s inference costs, we would like to note that for our experimental setting on A100 GPUs, performing inference on the full Imagenet-1K validation set (50K samples) takes \~12 seconds wall-clock time with ViT/B and \~16 seconds wall-clock time with ALViT/B. There is thus an overhead, though it follows an **evident sub-linear trend with respect to the total number of parameters**.
>
> \---
>
> **W4: Weak motivation for LLM use**
>
> As acknowledged by **reviewers BRBi, Ajvx and 4iid,** our work demonstrates that frozen LLM representations can significantly improve the quality of visual representations through improving robustness against adversarial background signals, even against significantly strong SSL baselines. Interestingly, as shown previously in LM4Vision \[A\], we show that this is achievable without utilizing any text-specific inputs for the LLM. We further demonstrate the exact effect of the LLM representations by utilizing a randomly-initialized LLM baseline in Table 2 of work, where we show that ALViT improves this baseline by **\~+0.4 across three random seeds.**
>
> Notably, we quantified this robustness by demonstrating significant performance gains of up to **\+2.1** in the Imagenet-9 adversarial backgrounds benchmark relative to an already strong MAE-pretrained ViT. Through our analysis in Section 5 and Appendix A, we showed that the attention gets more focused for the patches of informative foreground regions for ALViT compared to the ViT, evidenced by decreased attention entropy and increased patch norms.
>
> ---
>
> [F] Hou, Z., Sun, F., Chen, Y. K., Xie, Y., & Kung, S. Y. (2022). Milan: Masked image pretraining on language assisted representation. arXiv preprint arXiv:2208.06049.

---

> > ### Author Response · Authors · 2025-11-27
> > **Gentle Reminder for the Discussion Period**
> >
> > Dear Reviewer T4ti,
> >
> > Thank you for your review on our work!
> >
> > We believe that we addressed your primary concerns regarding our work by providing a detailed discussion on the novelty of our work with a highlight on our exact contributions, pointing to the quantitative implications of our analysis with attention entropy behaviors of ViT vs ALViT (with ALViT achieving **up to +2.1 gains over MAE ViT/B on Imagenet-9 adversarial backgrounds benchmark**), discussing the cost implications of our work and how ALViT achieves a more desirable trade-off with a comparable number of trainable parameters against similar baselines and finally quantitatively showing the necessity of frozen LLM representations in ALViT by highlighting its performance gains of **~+0.4 across three random seeds** against a randomly-initialized LLM baseline.
> >
> > Furthermore, we have shown that our methodology holds further promise for the future, with ALViT achieving **+7.0 improvement on kNN accuracy (81.0% vs 74.0%) and +3.5 improvement on linear probing accuracy (82.2% vs 78.7%)** over the MILAN ViT, a very strong SSL method for vision.
> >
> > We would thus appreciate it if you could please have a look at our replies and we are more than happy to get back to you if you have further questions or comments.
> >
> > Thank you very much again for your time and efforts on reviewing our paper!

---

### Official Review · Reviewer_4iid · 2025-10-26

**Soundness:** 3
**Presentation:** 3
**Contribution:** 2
**Rating:** 6
**Confidence:** 3

**Summary:**

This author introduces a novel framework called ALVIT, which aims to infuse the semantic knowledge of LLMs into self-supervised Vision Transformers to enhance performance on pure vision tasks. To address the modality mismatch between ViTs (vision-centric) and LLMs (text-centric), the authors propose a co-pretraining strategy. This strategy pre-trains the ViT backbone with a MAE objective, while simultaneously using the same MAE reconstruction loss to train Lora layers within the LLM fusion blocks. This joint optimization guides the ViT to produce LLM-friendly features and also enables the LLM to effectively interpret visual information. Experiments show that ALVIT significantly outperforms strong baselines (including MAE-ViT and LM4Vision) on ImageNet and its variants (such as IN-A, IN-C).

**Strengths:**

- **Co-Pretraining Strategy**: The core contribution of this paper is its co-pretraining framework. Using the MAE reconstruction loss to simultaneously guide the learning of the ViT backbone and the adaptation of LoRA layers in the LLM blocks is a novel and effective method to resolve the modality mismatch between visual and language representations.

- **Promising Performance**: ALVIT not only achieves performance improvements on ImageNet-1K but, more importantly, demonstrates significant advantages on multiple robustness benchmarks. This strongly proves that the model successfully leverages the knowledge from the LLM to enhance its resilience to out-of-distribution samples.

- **Solid Ablation Studies**: The authors validate the design choices of ALVIT through comprehensive ablation studies.

- **In-depth Mechanism Analysis**: Through a background robustness analysis on ImageNet and visualization of attention entropy, the paper provides profound insights into ALVIT's working mechanism. The analysis indicates that ALVIT exhibits a stronger ability to distinguish between background and foreground, which explains the source of its robustness.

**Weaknesses:**

- **Insufficient Discussion on MAE as an SSL Paradigm**: The authors chose MAE because its reconstruction loss is suitable for co-training. However, the authors' own analysis (Figure 3(a)) shows that the attention pattern of the MAE ViT/B baseline is "indifferent" to background and foreground regions. Given that other SSL paradigms (like DINO) are known for their strong foreground/background separation capabilities, the paper should further discuss why MAE is the sole or optimal choice for this task and whether other SSL objectives were considered.

- **Lack of Explanation for "Stronger LLMs Not Bringing Gains"**: A surprising finding (Table 3) is that using more advanced LLMs (like Gemma 2 or LLaMA 3.1) did not bring any performance improvement over the older LLaMA 1. This contradicts the intuition that stronger LLMs should provide richer semantic knowledge. The paper reports this phenomenon but fails to provide an in-depth discussion or hypothesis. Does this imply that ALVIT primarily utilizes the general transformer architecture of the LLM blocks rather than their specific, more advanced semantic knowledge?

- **Limited and Modest Gains on Downstream Tasks**: Although the paper includes object detection and instance segmentation results on MS COCO in Appendix B.1, these evaluations are not presented in the main text. Furthermore, while the results show consistent improvements, the gains are relatively modest (e.g., only a +0.5 increase in Bounding Box AP on COCO). To more comprehensively demonstrate that ALVIT is a superior representation learning method, the authors should present evaluations on a broader range of downstream tasks in the main paper.

**Questions:**

Please see the weakness section above.

---

> ### Author Response · Authors · 2025-11-21
> **Response for W1 (1/2)**
>
> We thank the reviewer for finding our method **novel and effective**, and finding our work to have **promising performance, solid ablation studies and in-depth mechanism analysis.** We also appreciate the reviewer’s constructive questions and feedback on comparisons with different SSL baselines, and our method’s behavior with stronger LLMs.
>
> Below, we present our responses to each of the issues raised by the reviewer.
>
> ---
>
> **W1. Insufficient discussion on MAE as an SSL paradigm**
>
> We chose MAE (He et al., CVPR 2022\) over other self-supervised methods for vision (Caron et al., CVPR 2021; He et al., CVPR 2020 ; Zhou et al., ICLR 2022\) due to its demonstrated strength in downstream vision-only recognition tasks, such as image classification, object detection and segmentation, especially following task-specific fine-tuning \[A, B\].
>
> We also agree with the reviewer that MAE underperforms these older baselines in linear separability as shown with linear probing and kNN performance, primarily due to its image-level reconstructive objective compared to these baselines’ contrastive objectives. This was also demonstrated and discussed in detail in the literature as a general difference between image-level reconstruction-based and contrastive SSL methods  \[C\]. Accordingly, we will incorporate a short discussion mentioning this design choice in our related works section.
>
> Furthermore, to solidify the benefits of ALViT on a broader range of SSL methods, below we present additional results with DINO \[D\] using the ViT/S configuration (100 epochs of pretraining) and and with MILAN \[E\], a stronger masked-image modeling approach which aims to reconstruct the CLIP visual encoder features instead of original image features as in MAE, outperforming MAE significantly with linear probing and kNN. For both of these SSL methods, we have integrated the ALViT in a similar manner to the MAE, and followed the exact set of hyperparameters configurations as the original works.
>
> The results below underscore the benefits of ALViT in a similar way: MILAN ALViT outperforms the vanilla MILAN ViT by **up to** **\+9.3 on kNN accuracy** while the DINO ALViT outperforms the DINO ViT **up to \+0.8 on kNN accuracy.**
>
> **Imagenet-1K Top-1 Accuracy with MILAN and DINO using kNN with different k values**
>
> | Model | k=10 | k=20 | k=100 | k=200 |
> | :---- | :---- | :---- | :---- | :---- |
> | MILAN ViT/B \[E\] | 74.0% | 73.9% | 72.0% | 70.7% |
> | MILAN ALViT/B (*ours*) | **81.0% (+7.0)** | **80.9% (+7.0)** | **80.4% (+8.4)** | **80.0% (+9.3)** |
> | DINO ViT/S \[D\] | 69.2% | 68.9% | 66.7% | 65.4% |
> | DINO ALViT/S (*ours*) | **69.7% (+0.5)** | **69.6% (+0.7)** | **67.4% (+0.7)** | **66.2% (+0.8)** |
>
> We believe that these results significantly improve our claims through quantitatively highlighting the benefits of ALViT on a wider range of SSL methods and thank the reviewer for their valuable comments.
>
> ---
>
> \[A\] Li, Y., Mao, H., Girshick, R., & He, K. (2022, October). Exploring plain vision transformer backbones for object detection. In European conference on computer vision (pp. 280-296). Cham: Springer Nature Switzerland.
>
> \[B\] He, K., Chen, X., Xie, S., Li, Y., Dollár, P., & Girshick, R. (2022). Masked autoencoders are scalable vision learners. In Proceedings of the IEEE/CVF conference on computer vision and pattern recognition (pp. 16000-16009).
>
> \[C\] Shekhar, S., Bordes, F., Vincent, P., & Morcos, A. (2022). Understanding contrastive versus reconstructive self-supervised learning of vision transformers. In NeurIPS 2022 Workshop: Self-Supervised Learning—Theory and Practice2022.
>
> \[D\] Caron, M., Touvron, H., Misra, I., Jégou, H., Mairal, J., Bojanowski, P., & Joulin, A. (2021). Emerging properties in self-supervised vision transformers. In Proceedings of the IEEE/CVF international conference on computer vision (pp. 9650-9660).
>
> \[E\] Hou, Z., Sun, F., Chen, Y. K., Xie, Y., & Kung, S. Y. (2022). Milan: Masked image pretraining on language assisted representation. arXiv preprint arXiv:2208.06049.

---

> ### Author Response · Authors · 2025-11-21
> **Response for W2 and W3 (2/2)**
>
> **W2. Lack of explanation for stronger LLMs not bringing gains**
>
> We would first like to note that ALViT’s gains are not due to the general transformer architecture of the LLM blocks and the pretrained knowledge of the LLMs are strictly necessary for achieving them. We demonstrated this crucial aspect of ALViT through an ablation with a randomly-initialized LLM block in our main paper **(Table 2, rows e & f)**, where we showed that **ALViT outperforms all similar baselines including the randomly-initialized LLM baseline by at least \+0.4.**
>
> Regarding the performance saturation with stronger LLMs, we hypothesize this is likely due to the inherent tendency of LLMs to develop converging representations towards their final layers, as they are all trained for the ultimate goal of next-token prediction. The knowledge relevant to *general semantic structure* for a language-aligned feature space might already be sufficiently encoded in LLaMA 1's final block, and subsequent, more advanced textual knowledge may not translate into further visual gains within the current MAE-based co-adaptation. Existing literature also supports this hypothesis to an extent, with some works showing that older and newer LLaMA variants \[F, G\] and Gemma variants \[F, H\] have convergent representations towards their final blocks.
>
> We would also like to note that the performance of stronger LLM variants could benefit further from more detailed hyperparameter tuning, such as those involving LoRA, where we could not perform due to computational restrictions during the limited time of the author response period.
>
> Accordingly, we believe that effectively leveraging the representations of these stronger LLMs is an interesting future work to explore and we sincerely thank the reviewer for bringing this up.
>
> ---
>
> **W3. Limited and modest gains on downstream tasks**
>
> As the reviewer stated, we have shown that our **ALViT consistently outperforms vanilla ViT in object detection and instance segmentation tasks in the MS COCO benchmark**. We will be moving these results to the main paper.
>
> We would also like to note that we provided additional results on the Imagenet-Segmentation benchmark in Appendix B.4, where we showed that ALViT outperforms the MAE ViT/B baseline by up to **\+1.5, \+1.4 and \+1.3 respectively on Imagenet-Segmentation-50, \-300 and \-919 benchmarks.** For further reference, below we present the full results for this benchmark along with the results for LM4Vision, where we show that ALViT further outperforms LM4Vision by up to **\+3.1, \+2.2 and \+2.0 respectively on Imagenet-Segmentation-50, \-300 and \-919 benchmarks.**
>
> **Imagenet-Segmentation IoU Results with Baselines and ALViT**
>
> | Dataset | IN Segmentation \- 50 |  | IN Segmentation \- 300 |  | IN Segmentation \- 919 |  |
> | :---- | :----: | :----: | :----: | :----: | :----: | :----: |
> | **Feature** | **Freq.** | **Mag.** | **Freq.** | **Mag.** | **Freq.** | **Mag.** |
> | LM4Vision  | 40.8 | 40.7 | 41.8 | 41.3 | 41.6 | 41.3 |
> | MAE ViT/B | 39.8 | 42.3 | 40.9 | 42.8 | 40.8 | 42.8 |
> | ALViT/B (*ours*) | **41.3** | **43.8** | **42.3** | **43.5** | **42.1** | **43.3** |
>
> Finally, to further solidify the benefits of ALViT, we performed supplementary experiments on DAVIS-2017 Video Object Segmentation \[I\] task following the existing evaluation protocol in the self-supervised learning literature \[D\]. The results below demonstrate that ALViT outperforms the LM4Vision baseline by **\+2.3 on mean region similarity with \+1.8 in overall performance** while outperforming the MAE ViT/B baseline by **\+1.7 in overall performance \+1.4 on mean region similarity.**
>
> **DAVIS-2017 Video Object Segmentation Results with Mean Region Similarity ($J\_M$) and Contour-based Accuracy ($F\_M$)**
>
> | Model | $(J$ \&  $F)\_M$ | $J\_M$ | $F\_M$ |
> | :---- | :---- | :---- | :---- |
> | LM4Vision ViT/B | 57.2 | 55.5 | 59.0 |
> | MAE ViT/B | 57.3 | 56.4 | 58.2 |
> | ALViT/B (*ours*) | **59.0 (+1.7)** | **57.8 (+1.4)** | **60.1 (+1.9)** |
>
> ---
> [D] Caron, M., Touvron, H., Misra, I., Jégou, H., Mairal, J., Bojanowski, P., & Joulin, A. (2021). Emerging properties in self-supervised vision transformers. In IEEE/CVF international conference on computer vision.
>
> [F] Huh, M., Cheung, B., Wang, T., & Isola, P. (2024, July). Position: The platonic representation hypothesis. In Forty-first International Conference on Machine Learning.
>
> [G] Lan, M., Torr, P., Meek, A., Khakzar, A., Krueger, D., & Barez, F. (2024). Quantifying Feature Space Universality Across Large Language Models via Sparse Autoencoders.
>
> [H] Rufail, A., Rathore, S., Son, D., Simon, A., Dave, S., Zhang, D., Blondin, C., O’Brien, S., & Zhu, K. (2025). Semantic convergence: Investigating shared representations across scaled LLMs. In Proceedings of the ACL 2025 Student Research Workshop.
>
> [I] Pont-Tuset, J., Perazzi, F., Caelles, S., Arbeláez, P., Sorkine-Hornung, A., & Van Gool, L. (2017). The 2017 davis challenge on video object segmentation.

---

> > ### Comment · Reviewer_4iid · 2025-11-23
> >
> > Thanks for the rebuttal. I am still interested in W1: why does the self-distilled SSL paradigm (DINO) underperform with ALViT significantly compared to MAE-based models? Can the authors provide some rationale for this phenomenon?

---

> ### Author Response · Authors · 2025-11-23
> **Further Response for W1**
>
> Thank you for going through with our response. We would like to clarify a potential misunderstanding and kindly ask for the reviewer's response in case we misinterpreted their question.
>
> The results reported with DINO above were for ViT/S and ALViT/S, as we could not pretrain a ViT/B or ALViT/B with DINO due to time and computational constraints around the rebuttal period. In terms of kNN accuracy, our findings actually indicate that DINO ViT/S outperforms MAE ViT/B, even with its smaller model size. These results are also consistent with the existing literature, as they also show the superiority of DINO over MAE on kNN evaluations.
>
> Furthermore, we also showed that MAE ALViT/B outperforms MAE ViT/B by **up to** \+**2.0% in kNN accuracy** and DINO ALViT/S outperforms DINO ViT/S **up to \+0.8 on kNN accuracy**, which further demonstrates the strengths and versatility of our methodology. For reference, below we present these results jointly:
>
> **Imagenet-1K Top-1 Accuracy with MAE and DINO using kNN with different k values**
>
> | Model | k=10 | k=20 | k=100 | k=200 |
> | :---- | :---- | :---- | :---- | :---- |
> | MAE ViT/B | 41.2% | 41.9% | 40.5% | 38.9% |
> | MAE ALViT/B (*ours*) | **42.8% (+1.6)** | **43.3 (+1.4)** | **42.3% (+1.8)** | **40.9% (+2.0)** |
> | DINO ViT/S | 69.2% | 68.9% | 66.7% | 65.4% |
> | DINO ALViT/S (*ours*) | **69.7% (+0.5)** | **69.6% (+0.7)** | **67.4% (+0.7)** | **66.2% (+0.8)** |
>
> Finally, we would like to note that the results presented in our main text (Tables 1, 2 and 3\) were all reported after end-to-end finetuning, following the original training recipe of MAE.
>
> Please let us know if we misinterpreted your question in any way. We are grateful for your time and thoughtful feedback\!

---

> > ### Author Response · Authors · 2025-11-23
> > **Brief Update on Further Response for W1**
> >
> > We have updated our response above to clarify the model sizes utilized in DINO (ViT/S and ALViT/S) versus MAE (ViT/B and ALViT/B). We are very grateful for your time and hope we were able to address your concerns!

---

> > > ### Comment · Reviewer_4iid · 2025-11-24
> > >
> > > Thank you for the prompt response. Based on the results provided above, I recommend that the authors include the experiments regarding different SSL models in the main paper. Furthermore, the performance gain of MILAN + ALViT is significantly higher than that of DINO and MAE. This highlights an interesting aspect of how different representations interact with LLMs. For now, I will maintain my original score, but I will review the other responses before making a final rating.

---

> > > > ### Author Response · Authors · 2025-11-24
> > > >
> > > > We thank the reviewer for their constructive feedback and we agree that incorporating these discussions on other SSL paradigms makes our work even stronger. We will include these results in the manuscript and thank the reviewer for their extremely helpful suggestions.
> > > >
> > > > Regarding MILAN’s impressive gains with ALViT, we hypothesize that our paradigm is particularly strong here as MILAN uses language-aligned CLIP visual encoder features as its reconstruction target. We will also include this discussion in the manuscript as suggested by the reviewer. Finally, we believe that this can be an exciting direction for future works to expand on.
> > > >
> > > > We are grateful for your prompt feedback and hope that we were able to address your concerns. Thank you for your time and consideration.

---

### Official Review · Reviewer_Ajvx · 2025-10-28

**Soundness:** 3
**Presentation:** 3
**Contribution:** 3
**Rating:** 6
**Confidence:** 4

**Summary:**

This paper introduces Adapted-Language Vision Transformers (ALViT), which integrate frozen LLM blocks into ViTs via a MAE + LoRA self-supervised training scheme. By co-adapting both modules, ALViT achieves higher accuracy than previous LLM-fusion baselines and robustness on ImageNet benchmarks and shows improved background sensitivity via attention-entropy analyses.

**Strengths:**

1. Clear and novel motivation: extend supervised only LM4Vision to SSL training.
2. Efficient tuning: The author find the frozen LM layer is not suitable for SSL training, and uses LoRA for finetuning without increasing much trainable parameters.
3. Solid empirical performance: although marginal, consistently outperforms MAE-ViT basleine and LM4Vision; robustness improvements are also convincing.
4. Thorough ablations: analyzes LoRA, parameter count, LLM layers, random initialization, and multiple seeds.
5. attention entropy visualizations support the hypothesis of improved information filtering and robustness.

**Weaknesses:**

1. Lack of justification for the training objective:
Masked Image Modeling is a well-established self-supervised learning objective, but it is no longer the most advanced one. Methods such as MoCo, DINO, and iBOT all outperform MAE by a large margin. Why do the authors choose MAE instead of adopting these stronger SSL objectives?

2. Lack of metrics:
The paper mainly evaluates fine-tuning accuracy. However, for SSL models, other important metrics include linear probing accuracy and kNN accuracy. Can ALViT also outperform the MAE baselines on these metrics?

3. Missing baseline:
In Table 3, the authors compare different variants with roughly the same number of trainable parameters. However, an important baseline seems missing: simply increasing the depth of the MAE ViT-B to match the parameter count (for example, using a 13-layer ViT-B). Additionally, the paper primarily focuses on ViT-B-sized models—can ALViT maintain its advantage across larger model scales?

**Questions:**

1. Why do the authors choose Masked Image Modeling as the training objective, given that more advanced SSL methods such as MoCo, DINO, and iBOT have been shown to outperform MAE by a large margin?

2. Can the authors report additional SSL metrics such as linear probing accuracy and kNN accuracy to evaluate whether ALViT also outperforms MAE baselines on these measures?

3. Can the author provide an additional baseline where the depth of MAE ViT-B is simply increased (e.g., using a 13-layer ViT-B) to match the trainable parameter count of ALViT?

4. Can the authors verify whether ALViT maintains its performance advantages across different model scales beyond ViT-B?

Since there is no borderline this year, I would still recommend borderline accept at this point for the clear motivation, solid results and experiments.

---

> ### Author Response · Authors · 2025-11-21
> **Response for W1 (1/2)**
>
> We thank the reviewer for finding our work to have **clear and novel motivation, solid empirical performance, thorough ablations and strong visualizations.** We also thank the reviewer for their constructive questions and feedback on comparisons with different SSL baselines and with different evaluation methods.
>
> Below, we present our responses to each of the issues raised by the reviewer.
>
> ---
>
> **W1. Lack of justification on the training objective**
>
> We chose MAE (He et al., CVPR 2022\) over other self-supervised methods for vision (Caron et al., CVPR 2021; He et al., CVPR 2020 ; Zhou et al., ICLR 2022\) due to its demonstrated strength in downstream vision-only recognition tasks, such as image classification, object detection and segmentation, especially following task-specific fine-tuning \[A, B\].
>
> We also agree with the reviewer that MAE underperforms these older baselines with linear probing and kNN, primarily due to its image-level reconstructive objective compared to these baselines’ contrastive objectives. This was also demonstrated and discussed in detail in the literature as a general difference between image-level reconstruction-based and contrastive SSL methods  \[C\]. Accordingly, we will incorporate a short discussion mentioning this design choice in our related works section.
>
> Furthermore, to solidify the benefits of ALViT on a broader range of SSL methods, below we present additional results with DINO \[D\] using the ViT/S configuration (100 epochs of pretraining) and and with MILAN \[E\], a stronger masked-image modeling approach which aims to reconstruct the CLIP visual encoder features instead of original image features as in MAE, outperforming MAE significantly with linear probing and kNN. For both of these SSL methods, we have integrated the ALViT in a similar manner to the MAE, and followed the exact set of hyperparameters configurations as the original works.
>
> The results below underscore the benefits of ALViT in a similar way: MILAN ALViT outperforms the vanilla MILAN ViT by **up to** **\+9.3 on kNN accuracy** while the DINO ALViT outperforms the DINO ViT **up to \+0.8 on kNN accuracy.**
>
> **Imagenet-1K Top-1 Accuracy with MILAN and DINO using kNN with different k values**
>
> | Model | k=10 | k=20 | k=100 | k=200 |
> | :---- | :---- | :---- | :---- | :---- |
> | MILAN ViT/B \[E\] | 74.0% | 73.9% | 72.0% | 70.7% |
> | MILAN ALViT/B (*ours*) | **81.0% (+7.0)** | **80.9% (+7.0)** | **80.4% (+8.4)** | **80.0% (+9.3)** |
> | DINO ViT/S \[D\] | 69.2% | 68.9% | 66.7% | 65.4% |
> | DINO ALViT/S (*ours*) | **69.7% (+0.5)** | **69.6% (+0.7)** | **67.4% (+0.7)** | **66.2% (+0.8)** |
>
> We believe that these results significantly improve our claims through quantitatively highlighting the benefits of ALViT on a wider range of SSL methods and thank the reviewer for their valuable comments.
>
> ---
>
> \[A\] Li, Y., Mao, H., Girshick, R., & He, K. (2022, October). Exploring plain vision transformer backbones for object detection. In European conference on computer vision (pp. 280-296). Cham: Springer Nature Switzerland.
>
> \[B\] He, K., Chen, X., Xie, S., Li, Y., Dollár, P., & Girshick, R. (2022). Masked autoencoders are scalable vision learners. In Proceedings of the IEEE/CVF conference on computer vision and pattern recognition (pp. 16000-16009).
>
> \[C\] Shekhar, S., Bordes, F., Vincent, P., & Morcos, A. (2022). Understanding contrastive versus reconstructive self-supervised learning of vision transformers. In NeurIPS 2022 Workshop: Self-Supervised Learning—Theory and Practice2022.
>
> \[D\] Caron, M., Touvron, H., Misra, I., Jégou, H., Mairal, J., Bojanowski, P., & Joulin, A. (2021). Emerging properties in self-supervised vision transformers. In Proceedings of the IEEE/CVF international conference on computer vision (pp. 9650-9660).
>
> \[E\] Hou, Z., Sun, F., Chen, Y. K., Xie, Y., & Kung, S. Y. (2022). Milan: Masked image pretraining on language assisted representation. arXiv preprint arXiv:2208.06049.

---

> ### Author Response · Authors · 2025-11-21
> **Response for W2 and W3 (2/2)**
>
> **W2. Lack of metrics**
>
> Following the reviewer’s question, below we report the linear probing and kNN accuracy of both our ALViT and MAE ViT/B baseline. These results follow a similar trend: ALViT outperforms the MAE ViT/B by **up to** \+**2.0% in kNN accuracy** and performs similarly well with MAE ViT/B in linear probing accuracy.
>
> **Imagenet-1K Top-1 Accuracy with Linear Probing and kNN Features with MAE**
>
> | Model | Linear Probing | kNN (k=10) | kNN (k=20) | kNN (k=100) | kNN (k=200) |
> | :---- | :---- | :---- | :---- | :---- | :---- |
> | MAE ViT/B | 65.0% | 41.2% | 41.9% | 40.5% | 38.9% |
> | ALViT/B (*ours*) | **65.1% (+0.1)** | **42.8% (+1.6)** | **43.3 (+1.4)** | **42.3% (+1.8)** | **40.9% (+2.0)** |
>
> For training the linear layer, we directly adhered to the hyperparameter configuration described in the original MAE work. We could not perform extensive hyperparameter tuning due to time and computational constraints contrary to the common practice for linear probing experiments in the field. We believe that the gains of ALViT could become more pronounced in linear probing experiments as well, as demonstrated by its kNN performance gains over the MAE ViT/B baseline.
>
> ---
>
> **W3. Missing baselines**
>
> We thank the reviewer for pointing out this potential baseline. We initially did not consider the baseline as having 13 blocks would introduce more trainable parameters than both ALViT and our baselines. We also agree with the reviewer that this baseline is important and started training this baseline with our standard training configuration. Due to time and computational constraints, we have not been able to obtain the final evaluation results yet, though we are doing our best to deliver them as soon as possible.
>
> Finally, we are also pretraining MAE ViT/L and ALViT ViT/L and have not been able to conclude them due to time and computational constraints. Likewise with the 13 blocks baseline, we are doing our best to deliver these results as soon as possible.

---

> > ### Comment · Reviewer_Ajvx · 2025-11-24
> > **Response to authors**
> >
> > I thank the authors for the timely response and the additional experiments. The results are strong and convincing. Since some experiments are still ongoing (e.g., the 13-block ViT-B), I will keep my current rating for now and revisit afterwards.

---

> > > ### Author Response · Authors · 2025-11-27
> > > **Response for the Remaining Experiments**
> > >
> > > We thank the reviewer for finding the **improvements of ALViT on a broader range of SSL methods to be strong and convincing.** We also thank the reviewer for their patience regarding the 13-block ViT/B baseline and ViT/L experiments.
> > >
> > > Below, we present the results for these remaining experiments. For the ViT/L comparison, we pretrained both the MAE ViT/L and ALViT/L models for 200 epochs due to the time constraints of the rebuttal period. For fine-tuning, we adhered to the standard MAE settings.
> > >
> > > ---
> > >
> > > **1\. 13-Block Baseline**
> > >
> > > Comparisons against the 13-block ViT baseline demonstrate that ALViT/B outperforms the 13-block ViT/B by **up to \+1.2% on kNN accuracy** and **\+0.3% on fine-tuning accuracy.** Crucially, ALViT/B achieves these gains despite having **\~500K fewer trainable parameters** than the 13-block baseline, confirming that our gains stem from the synergistic integration of the LLM rather than simply increased depth or capacity.
> > >
> > > **Imagenet-1K Top-1 Accuracy with Fine-tuning and kNN Features with MAE \- Base ViT**
> > >
> > > | Model | Fine-tuning | kNN (k=10) | kNN (k=20) | kNN (k=100) | kNN (k=200) |
> > > | :---- | :---- | :---- | :---- | :---- | :---- |
> > > | MAE ViT/B (12 Blocks) | 83.1% | 41.2% | 41.9% | 40.5% | 38.9% |
> > > | MAE ViT/B (13 Blocks) | 83.3% | 42.0% | 42.6% | 41.1% | 39.7% |
> > > | ALViT/B (*ours*) | **83.6% (+0.3)** | **42.8% (+0.8)** | **43.3 (+0.7)** | **42.3% (+1.2)** | **40.9% (+1.2)** |
> > >
> > > ---
> > > **2\. Scalability (ViT-L)**
> > >
> > > We observe consistent improvements with ALViT at the large model scale. ALViT/L outperforms the vanilla ViT/L by **\+0.5% in fine-tuning accuracy and up to \+1.0% in kNN accuracy.**
> > >
> > > **Imagenet-1K Top-1 Accuracy with Fine-tuning and kNN Features with MAE \- Large ViT (200 Epoch Pre-training)**
> > >
> > > | Model | Fine-tuning | kNN (k=10) | kNN (k=20) | kNN (k=100) | kNN (k=200) |
> > > | :---- | :---- | :---- | :---- | :---- | :---- |
> > > | MAE ViT/L | 83.7% | 39.7% | 40.3% | 39.0% | 37.5% |
> > > | ALViT/L (*ours*) | **84.2% (+0.5)** | **40.4% (+0.7)** | **41.3% (+1.0)** | **39.9% (+0.9)** | **38.4% (+0.9)** |
> > >
> > > We thank the reviewer for suggesting these meaningful baselines. Their inclusion has strengthened our primary claims by demonstrating the versatility and scalability of ALViT compared to deeper and larger baselines. We hope we have addressed the remaining questions of the reviewer and strengthened their confidence in our work.

---

### Official Review · Reviewer_BRBi · 2025-11-01

**Soundness:** 2
**Presentation:** 3
**Contribution:** 2
**Rating:** 4
**Confidence:** 5

**Summary:**

This paper, building on the foundation of LLM4Vision, introduces a new method for using LLMs to enhance vision-only capabilities. The main change is the addition of the Masked Auto-Encoding (MAE) pre-training task, during which the LLM is fine-tuned via LoRA. Overall, it has achieved a few improvements in image classification tasks. This work has inspirational significance (or heuristic value) for the study of vision-only pre-training paradigms that can simultaneously fine-tune both the visual encoder and the LLM.

**Strengths:**

1.  The paper is well-written, clearly articulated, and the methodology is relatively easy to follow.
2.  The exploration of applying LLMs within a vision-only pre-training paradigm is insightful.
3.  The feature analysis within the ablation study is well-executed, providing an intuitive visualization of the differences at the feature representation level resulting from the proposed training strategy.

**Weaknesses:**

1.  This work offers limited novelty built upon the LLM4Vision foundation. The finding that fine-tuning a few parameters on an adapted pre-training task can boost performance is rather straightforward and somewhat anticipated.
2.  Furthermore, judging from the final results, the performance improvement appears to be quite marginal.
3.  Regarding performance validation, the authors have primarily focused on classification tasks. As a general-purpose visual encoder, its effectiveness must be validated on a broader range of vision tasks (e.g., detection, segmentation, visual understanding).
4.  Moreover, the paper lacks direct comparisons with LLM4Vision across this wider set of tasks, making it difficult to fully assess the benefits of the proposed modifications.

**Questions:**

1.  What is the model's performance on a more diverse set of vision tasks (e.g., object detection, semantic segmentation)?
2.  Are there more experimental results providing a direct comparison against LLM4Vision under identical settings?

---

> ### Author Response · Authors · 2025-11-21
> **Response for W1 and W2 (1/2)**
>
> We thank the reviewer for finding that our work has **inspirational significance,** proposes an **insightful paradigm,** and contains **well-executed ablations and feature analyses.** We also appreciate the reviewer’s constructive questions and feedback on additional experiments with different vision tasks.
>
> Below, we present our responses to each of the issues raised by the reviewer.
>
> ---
>
> **W1: Limited novelty and anticipated results**
>
> As acknowledged by **reviewers Ajvx and 4iid,** our primary contribution is that we propose an effective visual backbone by successfully combining the powerful self-supervised MAE pretraining with the frozen knowledge of an LLM. We show that our fusion leads to robust gains on numerous challenging benchmarks (up to **\+0.5** on Imagenet-1K, **\+2.2** on Imagenet-A and up to **\+2.1** on Imagenet-9). Unlike LM4Vision \[A\], we show that simply inserting an LLM block does not help a strong MAE-pretrained backbone; the remarkable gains appear only when the block is equipped with lightweight LoRA adaptations. We also strengthen our experimental results with intriguing analyses on *why* our method works by analyzing *how* the ViT leverages the inherited LLM representations.
>
> Note that MAE is a very strong self-supervised pretraining strategy greatly improving over many self-supervised (e.g. BeiT \[B\]) and supervised baselines especially for dense prediction tasks like object detection \[C\]. Importantly, supervised-only pretrained backbones often result in sub-optimal performance, even against randomly-initialized visual backbones \[C\].
>
> Finally, we would like to note that ALViT is a more performant ViT backbone that could provide an improved starting point for more recent state-of-the-art masked-image modeling based approaches for computer vision, such as EVA-variants \[D, E\]. Therefore, we believe that ALViT is an interesting, novel idea for attaining a better alternative to vanilla ViT.
>
> ---
>
> **W2: Marginal performance improvements**
>
> The performance improvements brought in by ALViT are statistically significant and are not trivial, as demonstrated by the standard deviation of the performance across numerous Imagenet benchmarks in Table 1\. Throughout our experiments with Imagenet-1K, Imagenet robustness, Imagenet-Segmentation and MS COCO benchmarks, ALViT has shown consistent and steady improvement over all of our baselines across different vision tasks, achieving up to **\+0.5** on Imagenet-1K, **\+2.2** on Imagenet-A, **\+2.1** on Imagenet-9, up to **\+1.5** on Imagenet-Segmentation and **\+0.5** on MS COCO object detection task.
>
> To further solidify our results and demonstrate the strengths of ALViT, we also performed experiments with MILAN \[F\], a stronger masked-image modeling approach which aims to reconstruct the CLIP visual encoder features instead of original image features as in MAE. The results we obtained show that ALViT achieves **\+7.0 improvement on kNN accuracy (81.0% vs 74.0%) and \+3.5 improvement on linear probing accuracy (82.2% vs 78.7%)** over the vanilla MILAN ViT.
>
> Accordingly, we believe that our method has significant potential for future applications under an even wider range of applications.
>
> ---
>
> \[A\] Pang, Z., Xie, Z., Man, Y., & Wang, Y. X. (2023). Frozen transformers in language models are effective visual encoder layers. arXiv preprint arXiv:2310.12973.
>
> \[B\] Wang, Wenhui, et al. "Image as a foreign language: Beit pretraining for vision and vision-language tasks." Proceedings of the IEEE/CVF Conference on Computer Vision and Pattern Recognition. 2023\.
>
> \[C\] Li, Y., Mao, H., Girshick, R., & He, K. (2022, October). Exploring plain vision transformer backbones for object detection. In European conference on computer vision (pp. 280-296). Cham: Springer Nature Switzerland.
>
> \[D\] Fang, Y., Wang, W., Xie, B., Sun, Q., Wu, L., Wang, X., ... & Cao, Y. (2023). Eva: Exploring the limits of masked visual representation learning at scale. In Proceedings of the IEEE/CVF conference on computer vision and pattern recognition (pp. 19358-19369).
>
> \[E\] Fang, Y., Sun, Q., Wang, X., Huang, T., Wang, X., & Cao, Y. (2024). Eva-02: A visual representation for neon genesis. Image and Vision Computing, 149, 105171\.

---

> ### Author Response · Authors · 2025-11-21
> **Response for W3 and W4 (2/2)**
>
> **W3 & 4: Broader range of vision tasks and comparisons to LM4Vision**
>
> In Appendix B.1 and B.4, we have shown that ALViT outperforms the MAE ViT baseline by **\+1.5** on Imagenet-Segmentation and **\+0.5** on MS COCO object detection task. Furthermore, following the reviewer’s suggestion, we benchmarked LM4Vision in the Imagenet-Segmentation benchmark and report the results below, where we show that ALViT outperforms the LM4Vision baseline by up to **\+2.9, \+2.2 and \+2.0 respectively on** Imagenet-Segmentation-50, \-300 and \-919 benchmarks.
>
> **Imagenet-Segmentation IoU Results with Baselines and ALViT**
>
> | Dataset | IN Segmentation \- 50 |  | IN Segmentation \- 300 |  | IN Segmentation \- 919 |  |
> | :---- | :----: | :----: | :----: | :----: | :----: | :----: |
> | **Feature** | **Freq.** | **Mag.** | **Freq.** | **Mag.** | **Freq.** | **Mag.** |
> | LM4Vision  | 40.8 | 40.7 | 41.8 | 41.3 | 41.6 | 41.3 |
> | MAE ViT/B | 39.8 | 42.3 | 40.9 | 42.8 | 40.8 | 42.8 |
> | ALViT/B (*ours*) | **41.3** | **43.8** | **42.3** | **43.5** | **42.1** | **43.3** |
>
> We will be moving these results to the main paper.
>
> Finally, to further solidify the benefits of ALViT, we performed supplementary experiments on DAVIS-2017 Video Object Segmentation \[G\] task following the existing evaluation protocol in the self-supervised learning literature \[H\]. The results below demonstrate that ALViT outperforms the LM4Vision baseline by **\+2.3 on mean region similarity with \+1.8 in overall performance** while outperforming the MAE ViT/B baseline by **\+1.7 in overall performance.**
>
> **DAVIS-2017 Video Object Segmentation Results with Mean Region Similarity ($J\_M$) and Contour-based Accuracy ($F\_M$)**
>
> | Model | $(J$ \&  $F)\_M$ | $J\_M$ | $F\_M$ |
> | :---- | :---- | :---- | :---- |
> | LM4Vision ViT/B | 57.2 | 55.5 | 59.0 |
> | MAE ViT/B | 57.3 | 56.4 | 58.2 |
> | ALViT/B (*ours*) | **59.0 (+1.7)** | **57.8 (+1.4)** | **60.1 (+1.9)** |
>
> ---
>
> \[A\] Pang, Z., Xie, Z., Man, Y., & Wang, Y. X. (2023). Frozen transformers in language models are effective visual encoder layers. arXiv preprint arXiv:2310.12973.
>
> \[B\] Wang, Wenhui, et al. "Image as a foreign language: Beit pretraining for vision and vision-language tasks." Proceedings of the IEEE/CVF Conference on Computer Vision and Pattern Recognition. 2023\.
>
> \[C\] Li, Y., Mao, H., Girshick, R., & He, K. (2022, October). Exploring plain vision transformer backbones for object detection. In European conference on computer vision (pp. 280-296). Cham: Springer Nature Switzerland.
>
> \[D\] Fang, Y., Wang, W., Xie, B., Sun, Q., Wu, L., Wang, X., ... & Cao, Y. (2023). Eva: Exploring the limits of masked visual representation learning at scale. In Proceedings of the IEEE/CVF conference on computer vision and pattern recognition (pp. 19358-19369).
>
> \[E\] Fang, Y., Sun, Q., Wang, X., Huang, T., Wang, X., & Cao, Y. (2024). Eva-02: A visual representation for neon genesis. Image and Vision Computing, 149, 105171\.
>
> \[F\] Hou, Z., Sun, F., Chen, Y. K., Xie, Y., & Kung, S. Y. (2022). Milan: Masked image pretraining on language assisted representation. arXiv preprint arXiv:2208.06049.
>
> \[G\] Pont-Tuset, J., Perazzi, F., Caelles, S., Arbeláez, P., Sorkine-Hornung, A., & Van Gool, L. (2017). The 2017 davis challenge on video object segmentation. arXiv preprint arXiv:1704.00675.
>
> \[H\] Caron, M., Touvron, H., Misra, I., Jégou, H., Mairal, J., Bojanowski, P., & Joulin, A. (2021). Emerging properties in self-supervised vision transformers. In Proceedings of the IEEE/CVF international conference on computer vision (pp. 9650-9660).

---

> > ### Author Response · Authors · 2025-11-27
> > **Gentle Reminder for the Discussion Period**
> >
> > Dear Reviewer BRBi,
> >
> > Thank you for your review on our work!
> >
> > We believe that we addressed your primary concerns regarding our work by providing a detailed discussion on the novelty of our work with a highlight on our exact contributions, demonstrating the strengths of our method on a broader range of SSL methods (**achieving +7.0 improvement on kNN accuracy** over a very strong SSL baseline of MILAN ViT) and by highlighting our gains over LM4Vision on a broader range of vision tasks (**+2.9, +2.2 and +2.0 respectively on Imagenet-Segmentation-50, -300 and -919 benchmarks** and **+2.3 on mean region similarity with +1.8 in overall performance in DAVIS-2017 Video Object Segmentation benchmark**).
> >
> > We would appreciate it if you could please have a look at our replies and we are more than happy to get back to you if you have further questions or comments.
> >
> > Thank you very much again for your time and efforts on reviewing our paper!

---

### Author Response · Authors · 2025-11-21
**Response to All Reviewers**

We thank all reviewers for their time and efforts in going through our work and their thoughtful feedback. We are delighted that you found our motivation and methodology insightful, novel and interesting (**BRBi, Ajvx, 4iid**), our experiments and ablations solid and thorough (**Ajvx, 4iid**) and our paper to be well-written and well-organized (**BRBi, T4ti**).

Our paper proposes ALViT, an effective visual backbone by successfully combining the powerful self-supervised MAE pretraining with the frozen knowledge of an LLM through a joint optimization strategy. This joint optimization guides the ViT to produce LLM-aligned features and the LLM to effectively interpret visual information. Powered by this strategy, ALViT achieves significant performance gains on image classification (up to **\+0.5** on Imagenet-1K, **\+2.2** on Imagenet-A and up to **\+2.1** on Imagenet-9) and fine-grained visual recognition tasks (**\+1.7** on DAVIS-2017 Video Object Segmentation, up to **\+1.5** on Imagenet-Segmentation and **\+0.5** on MS COCO object detection). Additionally, we provide intriguing analysis on the attention entropy patterns of ALViT and demonstrate that ALViT achieves superior robustness against noisy background signals compared to a vanilla ViT.

Reviewers posed insightful questions and constructive comments, which we have addressed in detail in our individual responses.

---

### Author Response · Authors · 2025-12-03
**Rebuttal and Response Period Summary**

Dear Reviewers, AC, SAC, and PC,

We are grateful for your time, feedback, and professional oversight throughout the review and discussion phases. We thank the reviewers for recognizing:

* **Insightful, novel, and interesting methodology** (BRBi, Ajvx, 4iid)
* **Solid and convincing experiments** (Ajvx, 4iid)
* **In-depth feature-space and mechanism analyses** (BRBi, Ajvx, 4iid)
* **Clear and well-organized presentation** (BRBi, T4ti)

During the author response period, we substantially strengthened the paper with the following additions:

* **Broader SSL variants.** ALViT consistently improves across multiple SSL paradigms, achieving **up to \+9.3 kNN**with MILAN and **up to \+0.8 kNN** with DINO over their respective baselines. (BRBi, Ajvx, 4iid, T4ti)
* **More fine-grained benchmarks.** ALViT achieves **\+2.3** on DAVIS-2017 and **up to \+2.9** on ImageNet-Segmentation compared to LM4Vision. (BRBi, 4iid, T4ti)
* **Additional evaluation protocols.** ALViT improves MAE’s kNN accuracy by **up to \+2.0**, confirming robustness under alternative SSL metrics. (Ajvx, 4iid)
* **Stronger baselines and larger model scales.** ALViT outperforms a 13-block ViT/B (**up to \+1.2% kNN**) despite fewer trainable parameters, and improves ViT/L by **up to \+1.0% kNN**. (Ajvx)

We also clarified our contributions relative to LM4Vision, emphasized the **consistency and statistical significance** of ALViT’s gains across diverse benchmarks, and demonstrated its **general applicability across SSL methods, model scales, and downstream tasks**. These additions address concerns regarding novelty, SSL choices, empirical strength, and breadth of evaluation.

Reviewers Ajvx and 4iid found the new results **strong and convincing** and indicated they would revisit their ratings towards the end of the rebuttal period. While the other reviewers did not further respond during the discussion period, we believe that our extended evidence and clarifications directly address their core concerns regarding novelty, the strength and breadth of empirical gains, the choice of SSL objectives, and downstream applicability.

In summary, these updates substantially strengthen the paper and clarify the contributions and practical impact of ALViT. We sincerely thank the reviewers, AC, SAC, and PC for their careful assessments and constructive feedback.

---

### Meta-Review · Area_Chair_ejpu · 2026-01-09

**Summary:**

- Meta Review

This paper received borderline scores from the reviewers, with ratings reflecting mixed opinions on its contributions. After carefully considering all reviews and the author rebuttal, I recommend rejection for this submission.

- Summary of Reviews:

The reviewers raised several concerns, with the most significant ones centering on limited novelty and marginal performance improvements. Reviewer BRBi, in particular, provided a thorough critique highlighting that the proposed method does not introduce sufficiently novel insights or techniques beyond existing approaches.

- AC's Assessment:

I largely agree with Reviewer BRBi's assessment regarding the limited novelty and minor improvements. From my perspective, an even more fundamental concern lies in the practical utility of the proposed approach. The method introduces a large-scale LLM into the vision encoder training pipeline, which incurs substantial computational overhead. However, the resulting performance gains are modest and do not appear to justify this significant increase in resource requirements.
In my view, for a paper to merit acceptance, it should demonstrate strength in at least one of the following dimensions: (1) novelty—offering new insights, observations, or methodological contributions that advance our understanding of the problem; (2) empirical improvement—achieving substantial and consistent gains over existing methods; or (3) practical utility—providing an approach that is efficient, scalable, and readily deployable in real-world scenarios.
Unfortunately, the current submission falls short across all three criteria. The novelty is incremental, the improvements are marginal, and the computational cost of integrating LLMs into the training process undermines practical applicability. Given these limitations, I do not believe the paper meets the acceptance threshold for this venue.

Decision: Reject

**Reviewer Concerns:**

Addressed Concerns.

- Broader range of vision tasks and comparisons to LM4Vision
- Lack of justification on the training objective
- Insufficient discussion on MAE as an SSL paradigm
- Lack of metrics
- Missing baselines

Outstanding Concerns

- limited novelty and anticipated results
- Marginal performance improvements
- Weak motivation for LLM use

**Reviewer Scores:**

The paper received initial scores of 6, 6, 4, 4, placing it in the borderline range. Upon reviewing the content and feedback, it becomes evident that the method lacks simplicity, efficiency, and novelty—the performance improvements are modest relative to the substantial computational overhead introduced by LLM integration, and the approach offers limited inspirational value, resulting in constrained practical utility.

---

### Decision · Program_Chairs · 2026-01-26

Reject